# Bicomponent nano- and microfiber aerogels for effective management of junctional hemorrhage

S. M. Shatil Shahriar[1,2,8], Syed Muntazir Andrabi [1,8], Al-Murtadha Al-Gahmi[3,8], Zishuo Yan[1], Alec D. McCarthy[1], Chenlong Wang[1], Zakariya A. Yusuf[1], Navatha Shree Sharma[1], Milton E. Busquets[1,4], Mallory I. Nilles[1], Carlos Poblete Jara[5], Kai Yang[6], Mark A. Carlson [3] ✉ & Jingwei Xie [1,7] ✉

Managing junctional hemorrhage is challenging due to ineffective existing techniques, with the groin being the most common site, accounting for approximately 19.2% of potentially survivable field deaths. Here, we report a bicomponent nano- and microfiber aerogel (NMA) for injection into deep, narrow junctional wounds to effectively halt bleeding. The aerogel comprises intertwined poly(lactic acid) nanofibers and poly(ε-caprolactone) microfibers, with mechanical properties tunable through crosslinking. Optimized aerogels demonstrate improved resilience, toughness, and elasticity, enabling rapid re-expansion upon blood contact. They demonstrate superior blood absorption and clotting efficacy compared to commercial products (i.e., QuikClot® Combat Gauze and XStat®). Most importantly, in a lethal swine junctional wound model (Yorkshire swine, both male and female, $n = 5$), aerogel treatment achieved immediate hemostasis, a 100% survival rate, no rebleeding, hemodynamic stability, and stable coagulation, hematologic, and arterial blood gas testing.

Mortality rates linked to uncontrolled junctional hemorrhage (JH) have remained unchanged, emerging as a prominent cause of preventable deaths in remote civilian and military settings[1]. Recent conflicts have underscored the significance of addressing severe bleeding, particularly from deep wounds involving the femoral artery/vein or junctional areas (e.g., inguinal and axilla), which accounts for 19.2% of potentially avoidable deaths, particularly in the prehospital phase[2–4]. Notably, 87% of these fatalities occur within an hour of injury before reaching advanced medical care[5,6]. A primary prehospital treatment for JH is the XStat® device, which uses an expandable cellulose sponge coated with

chitosan[7]. While it is effective in many cases, its weak mechanical properties, slow expansion, and pressure application may not be optimal for situations where surgical facilities are hours away, as these could potentially result in continued bleeding. When XStat® is unavailable, QuikClot® Combat Gauze (QCG) is often used, but it is also ineffective in promptly stopping severe hemorrhage from deep and junctional wounds, leading to high mortality rates[8]. Various other innovations, such as the Pelvic-Binder®, Abdominal Aortic and Junctional Tourniquet (AAJT®), and iTClamp®, have been proposed[9]. However, their efficacy is limited by complications that require further

[1]Department of Surgery - Transplant and Mary & Dick Holland Regenerative Medicine Program, College of Medicine, University of Nebraska Medical Center, Omaha, NE, USA. [2]Eppley Institute for Research in Cancer and Allied Diseases, College of Medicine, University of Nebraska Medical Center, Omaha, NE, USA. [3]Department of Surgery - General Surgery, College of Medicine, University of Nebraska Medical Center, Omaha, NE, USA. [4]Pancreatic Cancer Center of Excellence, University of Nebraska Medical Center, Omaha, NE, USA. [5]Department of Surgery - Vascular Surgery, College of Medicine, University of Nebraska Medical Center, Omaha, NE, USA. [6]Department of Surgery - Plastic & Reconstructive Surgery, College of Medicine, University of Nebraska Medical Center, Omaha, NE, USA. [7]Department of Mechanical and Materials Engineering, University of Nebraska Lincoln, Lincoln, NE, USA. [8]These authors contributed equally: S. M. Shatil Shahriar, Syed Muntazir Andrabi, Al-Murtadha Al-Gahmi. ✉e-mail: macarlso@unmc.edu; jingwei.xie@unmc.edu

investigation to assess their potential in reducing mortality rates and other issues.

To improve the current treatment, recent pre-clinical studies have explored injectable and expansile cryogels, sponges, or foams, showing promise in rodent tail or liver injury models[10–19]. However, many potential treatments currently under development have one or more following characteristics that could affect their utilities, such as: (i) remaining in a wet state, which is unsuitable for blood concentration, and long-term storage, and may lead to fungal contamination; (ii) requiring the incorporation of clotting factors, necessitating special storage conditions that are impractical for battlefield settings, and risk degradation during transportation; (iii) exhibiting poor mechanical properties and slow expansion; (iv) lacking a safe and effective field delivery system; and (v) being non-resorbable, necessitating complete removal. Moreover, their efficacy in improving survival remains uncertain in inappropriate large animal models, with post-treatment rebleeding a common issue.

To overcome the abovementioned problems, we report a bicomponent nano- and microfiber aerogel (NMA) primarily composed of polylactic acid (PLA) nanofibers (NFs) and poly(ε-caprolactone) (PCL) microfibers (MFs) for the management of JH. This aerogel was fabricated using electrospinning, wet spinning, freeze casting, and cross-linking. We hypothesize that the unique structural design of the aerogel integrates complementary properties, enabling tunable mechanical properties, rapid shape recovery in blood, absorption, and coagulation activation, and providing a tamponade effect, thereby ensuring effective management of lethal JH. PCL and PLA are biodegradable, biocompatible polyesters with hydrophobic backbones lacking reactive groups[20]. They are primarily synthesized via ring-opening polymerization (ROP) of α-hydroxy acid monomers. The PCL MF component is designed to provide mechanical flexibility, resilience, adaptability, and improved elongation and impact toughness while promoting blood cell infiltration and enabling rapid shape recovery through the release of elastic energy upon contact with blood[21,22]. Meanwhile, the PLA NF components enhance the aerogel's structural integrity, rigidity, and shape retention, making it essential for maintaining mechanical stability under stress[23–25]. Due to its high absorption properties, PLA NF is intended to entrap platelets and clotting factors, enhancing the aggregation of red blood cells (RBCs). Additionally, the gelatin-coated NMA is expected to accelerate blood coagulation by activating coagulation factors and further enhance mechanical support[26,27]. The physical entanglement between NFs and MFs, combined with their chemical binding, is anticipated to render the aerogel mechanically resilient and flexible.

## Results

### Fabrication and structural characterizations of aerogels

The synthesis process of NF and MF aerogels (NMAs) involved a combination of electrospinning, wet spinning, fiber cutting, freeze-casting, and cross-linking techniques (Fig. 1a–c). Firstly, two-dimensional (2D) PLA NF mats were fabricated through electrospinning (Fig. 1a). These mats were then treated with air plasma and cryocut into short NFs. The short NFs were dispersed in water and homogenized to create a suspension with average diameters and lengths of $0.29 \pm 0.08\,\mu m$ and $96.50 \pm 24.62\,\mu m$, respectively (Supplementary Fig. 1a). Secondly, the short PCL MF suspension was produced by wet spinning, followed by plasma treatment, cutting, and homogenization (Fig. 1b). The resultant short MFs had average diameters and lengths of $0.016 \pm 0.007\,mm$ and $2.72 \pm 1.15\,mm$, respectively (Supplementary Fig. 1b). Finally, the short NF and MF dispersions and 1% gelatin granules were mixed and homogenized in a weight ratio of NF/MF at 50:50. This mixture was frozen in cylindrical molds at −80 °C and then freeze-dried to yield biocomponent aerogels. To enhance the stability and integrity of the aerogels, they were cross-linked using 2.5% glutaraldehyde vapor (GA) (Fig. 1c). The average

diameter and length of NMA were $0.82 \pm 0.096\,cm$ and $3.71 \pm 0.40\,cm$, respectively (Supplementary Fig. 2a, b). The compressed NMA pellets could be injected into the deep and narrow wounds, where they might re-expand to form a tamponade and stop the bleeding (Fig. 1d).

Figure 1e showed micro-computed tomography (micro-CT) images of the vertical and horizontal cross-sections of NF aerogels (NA), MF aerogels (MA), and NMA, indicating different porous characteristics among the aerogel samples (Supplementary Movies 1–3). NA exhibited the most minor porosity, followed by NMA and MA. The scanning electron microscopy (SEM) images unveiled distinct structural features of aerogels (Fig. 1f). NA exhibited a nest-like structure composed of PLA NFs and gelatin. MA comprised PCL MFs and gelatin nanofibrils. NMA consisted of entangled PCL MFs, PLA NFs, and gelatin nanofibrils formed through phase separation during freeze casting. However, distinguishing between gelatin nanofibrils and short NFs in SEM images posed a challenge due to their integration and similar appearance. The weights of NA, MA, and NMA were measured as $0.073 \pm 0.007\,g$, $0.098 \pm 0.008\,g$, and $0.085 \pm 0.011\,g$, respectively (Fig. 1g). In contrast, each pellet of XStat® weighed $0.19 \pm 0.005\,g$, which was at least two times heavier than any individual aerogel (Fig. 1g and Supplementary Movie 4). Similarly, the average NA, MA, and NMA volumes were approximately $3.14\,cm^3$, significantly higher than the volume of XStat® at $1.96\,cm^3$. This significant weight difference relative to their volume highlighted the aerogels' lightweight properties compared to XStat®. Similarly, as the weight decreased, their surface area increased (Fig. 1h). Regardless of their compositions, each aerogel variant demonstrated a statistically significant increase in surface area compared to the XStat®. NMA and NA showed higher porosity compared to MA and XStat® (Fig. 1i). Further Micro-CT analysis showed XStat® and NA had similar percentages of open pores ($71.01 \pm 13.63\%$ and $60.45 \pm 6.99\%$) and closed pores ($28.98 \pm 13.63\%$ and $39.54 \pm 6.99\%$), while MA and NMA displayed higher percentages of open pores ($93.91 \pm 2.40\%$ and $91.75 \pm 1.71\%$) and lower percentages of closed pores ($6.08 \pm 2.39\%$ and $8.24 \pm 1.71\%$) (Fig. 1j). Additionally, pore diameter analysis revealed values of $238.50 \pm 101.20\,\mu m$ and $185.77 \pm 66.98\,\mu m$ for MA and NMA, respectively, while NA and XStat® exhibited pore diameters of $16.90 \pm 7.96\,\mu m$ and $5.47 \pm 1.35\,\mu m$, respectively (Supplementary Fig. 3).

### Characterization of mechanical properties and simulations of stress and blood flow distributions and wall shear stress

We conducted compression tests on aerogel and XStat® samples after blood absorption under various compressive strains (Fig. 2a–c). Figure 2a indicated a correlation between compressive force and compressive strain. NMA exhibited significantly higher compressive force than MA, NA, and XStat®. Under 90% compressive strain, NMA required $15.10 \pm 0.85\,N$ of compressive force, which was higher than the forces for MA ($8.93 \pm 0.39\,N$), NA ($5.87 \pm 1.23\,N$), and XStat® ($5.19 \pm 0.22\,N$) (Fig. 2b). Notably, NMA displayed a higher specific elastic modulus than NA, MA, and XStat® across various compressive strains. It sustained stress levels of $0.38 \pm 0.047\,MPa\,cm^3\,g^{-1}$, $0.88 \pm 0.03\,MPa\,cm^3\,g^{-1}$, $3.09 \pm 0.29\,MPa\,cm^3\,g^{-1}$, $15.69 \pm 1.34\,MPa\,cm^3\,g^{-1}$, and $28.54 \pm 1.61\,MPa\,cm^3\,g^{-1}$ at compressive strains of 20%, 40%, 60%, 80%, and 90%, respectively (Fig. 2c). Supplementary Fig. 4 and 5 highlighted the exceptional resilience of NMA, which withstands stress levels of $0.10 \pm 0.01\,MPa$, $0.53 \pm 0.045\,MPa$, and $0.96 \pm 0.054\,MPa$ at compressive strains of 60%, 80%, and 90%, respectively. These values were significantly higher than the stress levels of $0.063 \pm 0.006\,MPa$, $0.37 \pm 0.035\,MPa$, and $0.66 \pm 0.029\,MPa$ observed for XStat®.

Each sample underwent five compression test cycles to assess compression resilience and shape recovery speed further. Each cycle included three steps at different compressive strains (e.g., 70%, 80%, and 90%). In each cycle, the sample was compressed at a rate of 1 mm/s to the desired strain, held at that strain for 5 s, and then released at

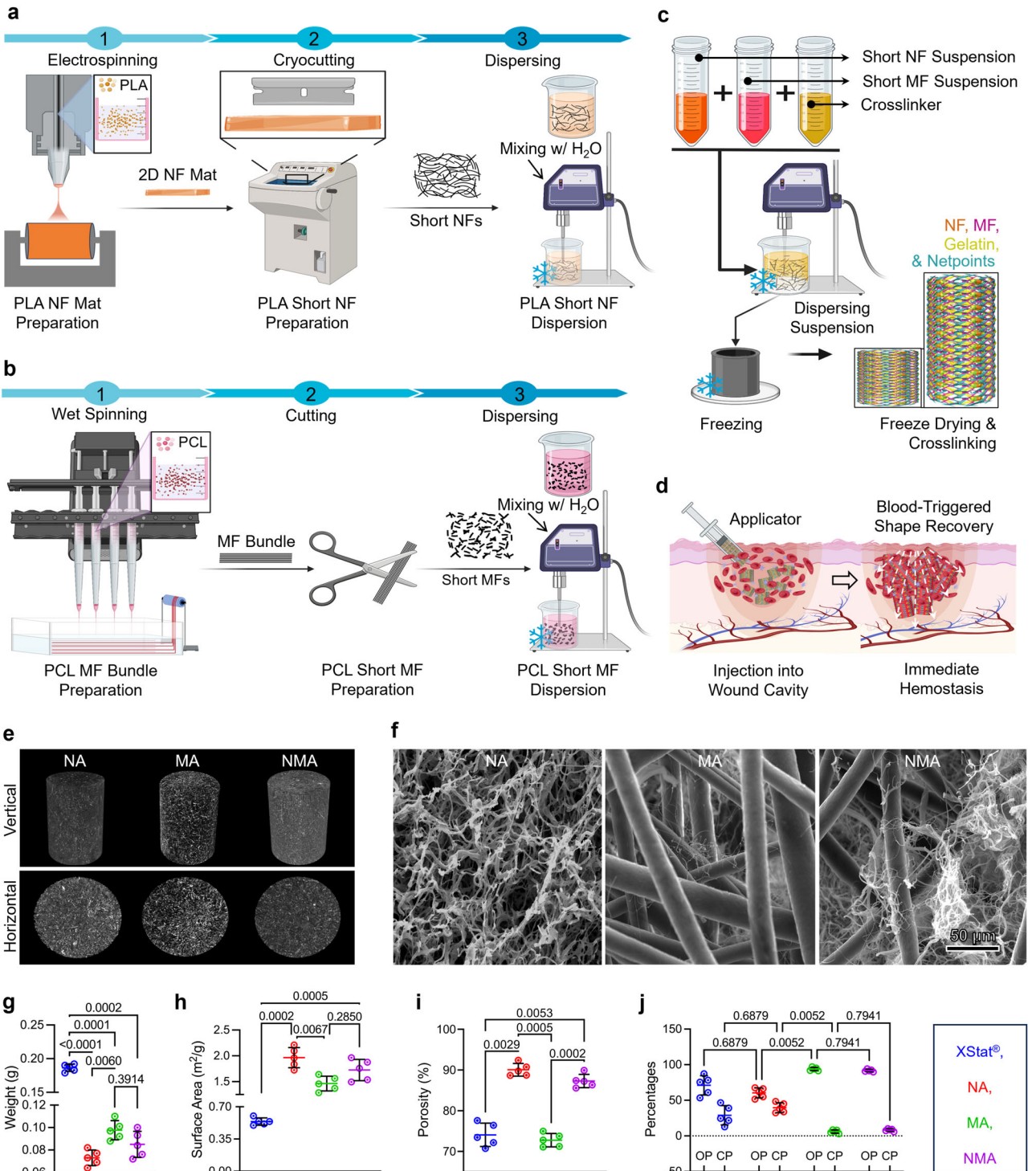

**Fig. 1 | Design, fabrication, and characterization of bicomponent nano- and microfiber aerogels. a–c** Illustration depicting the fabrication process of short nanofibers (NF) (**a**), short microfibers (MF) (**b**), and bicomponent nano- and microfiber aerogels (**c**). For NF fabrication: (1) NF mats produced by electrospinning, (2) short NFs produced by cryo-cutting 2D NF mats and air plasma treatment, and (3) NF dispersion. For MF fabrication: (1) MFs produced by wet spinning, (2) short MFs produced by cutting and air plasma treatment, and (3) MF dispersion. For NA fabrication: freeze casting of the short NF suspension containing gelatin as a cross-linker in a cylindrical mold followed by cross-linking with GA vapor and mold removal. For MA fabrication: freeze casting of the short MF suspension containing gelatin as a cross-linker in a cylindrical mold followed by cross-linking with GA vapor and mold removal. For NMA fabrication: freeze casting of the NF and MF suspension containing gelatin as a cross-linker in a cylindrical mold followed by cross-linking with GA vapor and mold removal. Schematics were created with BioRender.com. **d** Schematic illustrating the utilization of biocomponent nano- and microfiber aerogels as an injectable hemostatic material for the management of junctional hemorrhage. Created in BioRender. Shahriar, S. (2025) https://biorender.com/y89s148. **e** Micro-CT images displaying the vertical and horizontal cross sections of aerogels. **f** SEM images illustrating the morphology of the cross section of each sample Micrographs are representative of 3 independent samples. **g–j** The weight (**g**), surface area (**h**), porosity (**i**), and percentages of OP and CP (**j**) of different samples. OP and CP represent open and closed pores, respectively. Data were presented as mean ± s.d., *n* = 5 samples for (**g–j**). Statistical differences were determined using one-way (for **g–i**) or two-way (for **j**) ANOVA with Tukey's multiple comparisons test in GraphPad Prism. NF: nanofiber, MF: microfiber, NA: nanofiber aerogels, MA: microfiber aerogels, NMA: nano- and microfiber aerogels. Source data are provided as a Source data file.

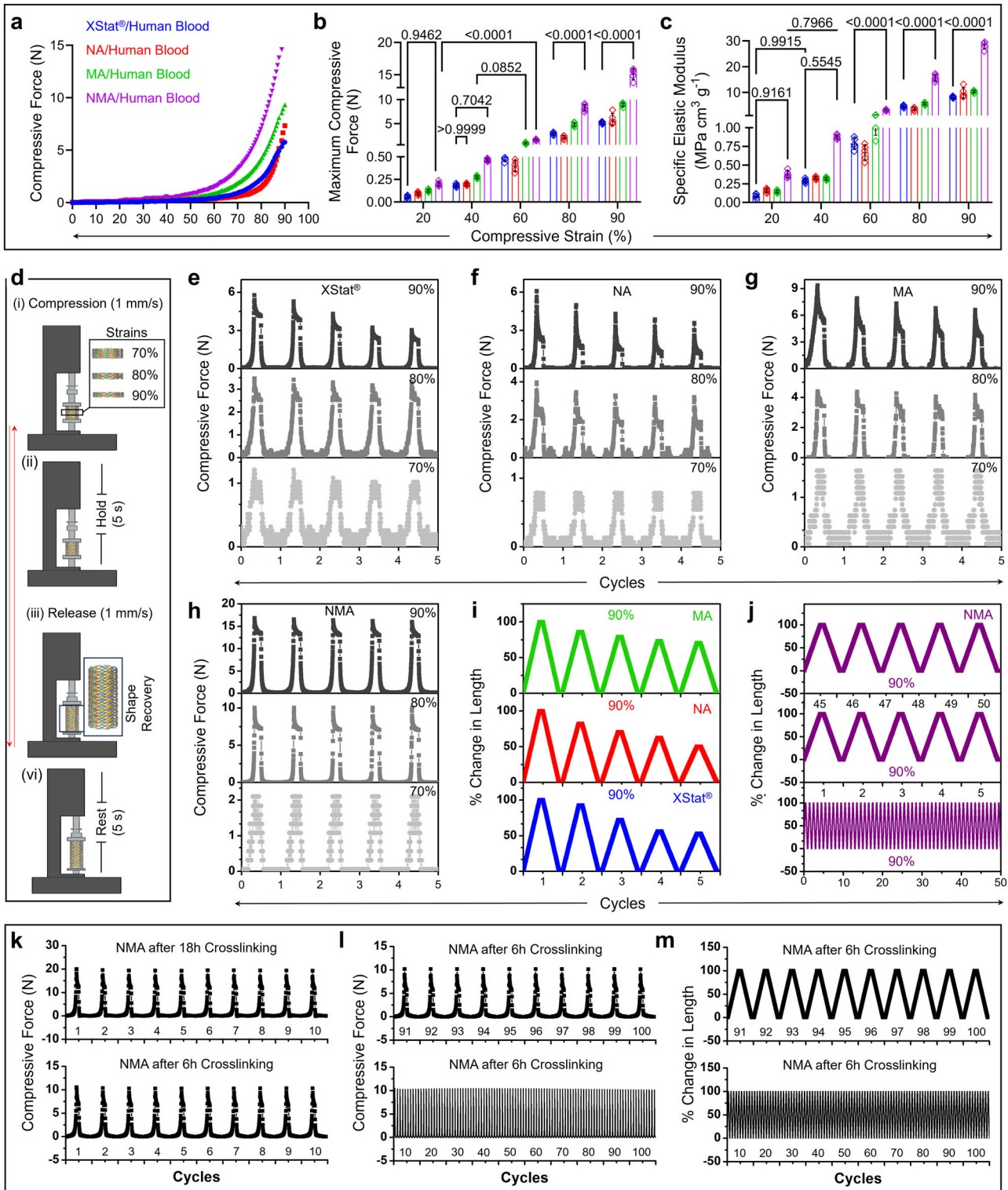

**Fig. 2 | Mechanical characterizations of bicomponent nano- and microfiber aerogels. a** The compressive stress-strain curves of XStat®, NA, MA, and NMA after absorbing human blood at 90% compressive strain. **b**, **c** Maximum compressive force, and specific elastic modulus at 20%, 40%, 60%, 80%, and 90% compressive strains. Values in **b** and **c** represent the mean ± s.d., ($n = 5$ samples). The significant difference was detected by two-way ANOVA with Tukey's multiple comparisons test. **d** Schematic illustration of the experimental setup for cyclic compression-relaxation tests of the tested samples in human blood. Created in BioRender. Shahriar, S. (2025)

https://BioRender.com/l59t519. **e**–**h** Cyclic compression test of XStat®, NA, MA, and NMA under 70%, 80%, and 90% compressive strains. **i**, **j** Changes in length of XStat®, NA, MA, and NMA during cyclic compression-relaxation test at 90% compressive strain. **k**, **l** The cyclic compression-relaxation curves and compressive force of NMA with different cross-linking times. **m** Changes in length of NMA with cross-linking for 6 h during 100 cycles of the compression-relaxation test at 90% compressive strain. NA: nanofiber aerogels. MA: microfiber aerogels, NMA: nano- and microfiber aerogels. Source data are provided as a Source data file.

1 mm/s to complete the cycle (Fig. 2d). This methodology could mimic the post-treatment external stress in the wound cavity caused by surrounding tissues. XStat® showed varied responses to different strains, demonstrating good resilience at 70% and 80% but decreased compressive resistance at 90% (Fig. 2e and Supplementary Fig. 6a). The compressive resistance dropped from $5.47 \pm 0.22$ N in the first cycle to $2.88 \pm 0.11$ N in the fifth cycle due to mechanical failure. Similarly, NA and MA exhibited shape recovery loss at 90% strain, decreasing from $5.80 \pm 0.38$ N and $9.14 \pm 0.26$ N in the first cycle to $3.36 \pm 0.26$ N and $6.23 \pm 0.30$ N, respectively (Fig. 2f, g and Supplementary Fig. 6b, c). This loss was likely due to the absence of a bi-continuous, entangled network. Interestingly, NMA displayed the best compression resilience and highest maximum compression force ($16.35 \pm 0.33$ N) without forming a mechanical fracture hysteresis loop (Fig. 2h and Supplementary Fig. 6d). All three aerogels and XStat® showed no apparent recovery loss when subjected to 70% strain, maintaining good shape and elasticity, indicating robust compression resilience under minor strains (Supplementary Fig. 7a–d). As the compressive strain increased from 70% to 80%, XStat® and NA exhibited escalating recovery loss of $17.04 \pm 2.58\%$ and $20.36 \pm 1.82\%$ in length from the first cycle to the fifth cycle, a scenario not observed in MA (Supplementary Figs. 6a–c and 7a–c). NMA showed robust mechanical properties without losing the total length at 70% and 80% strains (Supplementary Fig. 6d). Although NMA exhibited no deformation loss in total length, MA, NA, and XStat® showed $31.72 \pm 1.59\%$, $51.40 \pm 1.06\%$, and $49.21 \pm 1.63\%$ loss under 90% strain after five cycles (Fig. 2i). The compression resilience of NMA was further validated by the 50-cycle test, which showed no recovery loss in length from cycle 1 to cycle 50 (Fig. 2j). Figure 2k–m and Supplementary Fig. 8a, b demonstrated that mechanical strength could be finely tuned without compromising resilience by adjusting the cross-linking time. Reducing the cross-linking time from 18 h to 6 h decreased mechanical strength from $19.29 \pm 0.88$ N to $9.76 \pm 0.99$ N (Fig. 2k). Notably, NMA with a 6-h cross-linking time showed no hysteresis loop, mechanical failure, or loss in deformation and recovery over 100 testing cycles (Fig. 2l, m and Supplementary Fig. 8a, b).

Additionally, NMA exhibited the highest toughness among all tested samples, reaching a maximum value of $2061.00 \pm 68.47$ KJ/m³ (Fig. 3a). In comparison, MA and NA had $696.90 \pm 118.50$ and $101.10 \pm 14.26$ KJ/m³, respectively. XStat® showed the lowest toughness, with a value of $95.96 \pm 4.35$ KJ/m³ (Fig. 3a and Supplementary Fig. 9a). The high toughness of NMA allowed it to store a considerable amount of potential elastic energy of approximately $32.07 \pm 1.81$ KJ/m² before reaching plastic deformation, significantly surpassing other samples such as MA, NA, and XStat® (Fig. 3b and Supplementary Fig. 9b). Furthermore, despite its high toughness, NMA exhibited resilience characteristic, which was tenfold higher than that of XStat®. NMA showed "zero" mechanical failure in repeated cyclic tests over five cycles, while other tested samples exhibited a significant loss in resilience from cycle 1 to cycle 5 (Fig. 3c, and Supplementary Fig. 9b).

Next, finite element models were developed based on NA, MA, and NMA structures to understand the stress distribution and energy absorption mechanisms within aerogels. Figure 3d–f showed the stress distributions of these structures under external compressive forces along the Y-axis. The stress levels observed in NMA are notably lower than those in the mono-network counterparts. This reduction was attributed to the synergistic integration of MFs and NFs within NMA, facilitating efficient stress distribution and energy absorption, thereby mitigating stress concentrations during deformation (Supplementary Movie 5). The simulation results were consistent with the experimental findings.

Finally, two models representing NMA and XStat® were created based on their structural characteristics and used for fluid dynamic simulations of blood flow distribution to understand how structure affects blood absorption. NMA exhibited efficient blood flows, uniform distribution, consistent flow rates, and a well-aligned top-to-bottom flow direction, likely due to its larger porosity and higher percentage of open pores (Fig. 3g). In contrast, XStat® showed inefficient blood flows, uneven distributions, variable flow rates, and multiple flow directions, which may be attributed to its lower percentage of open pores and lower porosity (Fig. 3h). Additionally, NMA demonstrated uniform and relatively low wall shear stress, which may enhance effective blood absorption into the porous structure (Fig. 3i). Conversely, XStat® exhibited non-uniform and relatively high wall shear stress, potentially delaying blood absorption into the material (Fig. 3j).

## Characterization of water and blood absorption and shape recovery

NMA showed a significantly higher capacity for absorbing water and blood than XStat® (Fig. 4a, b). At early time points (e.g., 3 s and 5 s), NMA absorbed substantially more water and blood than XStat® (Fig. 4c, d). Additionally, NMA exhibited higher water and blood absorption rates (Fig. 4e, f). Unlike Surgicel®, which reduced the blood solution pH to approximately 2.830, aerogels showed no change in pH upon blood absorption (Supplementary Fig. 10)[4]. We further evaluated the fluid-triggered shape-recovery properties of NMA and XStat® (Supplementary Movies 6–9). All aerogel samples, when compressed into pellets in the dry state, could recover their original shapes upon absorbing water, achieving 100% shape recovery within 5 s (Supplementary Fig. 11). Supplementary Table 1 summarized the volumetric expansion ratios for different aerogels, ranging from $11.87 \pm 3.92$ for MA to $19.60 \pm 6.40$ for NA. XStat® exhibited a significantly lower expansion ratio of $6.01 \pm 0.37$, in contrast to the $15.72 \pm 0.56$ observed for NMA. NMA demonstrated a significantly shorter recovery time ($2.50 \pm 0.5$ s) and rate ($0.5 \pm 0.1$ s/cm) in water compared to XStat® (Fig. 4g, h). After absorbing blood, the compressed NA, MA, and NMA samples achieved 100% shape recovery within $6.67 \pm 2.08$ s, $6.00 \pm 1.00$ s, and $4.83 \pm 0.77$ s (Fig. 4i, j). However, XStat® remained compressed and could not recover further within 20 s. A 5.5 cm long NMA fully recovered its shape within 5 s in contact with blood, while a 3 cm long XStat® recovered only 0.4 cm within the same period due to a lack of proper structure (Supplementary Fig. 12a–c and Supplementary Movies 8, 9). The NMA exhibited complete wetting behavior with both water and human blood, showing a contact angle of zero and demonstrating rapid absorption capacity (Supplementary Movie 10).

In addition, the shape recovery time of NMA after absorbing blood was significantly shorter than that of reported shape-recoverable hemostats (Fig. 4k and Supplementary Table 2)[4,13,14,16,17,19,28–30]. Notably, many studies have shown that blood, due to its higher viscosity, tends to prolong the shape recovery time of hemostats compared to water[16,19]. However, for NMA, there was no significant difference in shape recovery rate between water and blood absorption (Fig. 4h, k and Supplementary Movies 7, 9). The wet aerogels were also compressible and capable of reabsorbing fluids (Supplementary Movie 11). The cross-sectional structures of various samples after compression and subsequent shape recovery upon water absorption were further examined using SEM (Fig. 4l). Under compression, the pores appeared deformed and condensed. Upon absorbing water, the deformed pores reverted to their original shapes without noticeable changes in size. Additionally, SEM images of the surfaces and cross-sections of samples were taken after blood absorption at different time points (Fig. 4m). A substantial number of blood cells aggregated on the surfaces of both NMA and XStat® at all the time points. The absorbed blood cells showed an uneven distribution on the cross-section of XStat®, whereas NA, MA, and NMA exhibited relatively uniform distributions. Most importantly, only a small number of RBCs aggregated on the cross-sections of XStat®, while a significantly higher number of RBCs accumulated on the cross-sections of NMA.

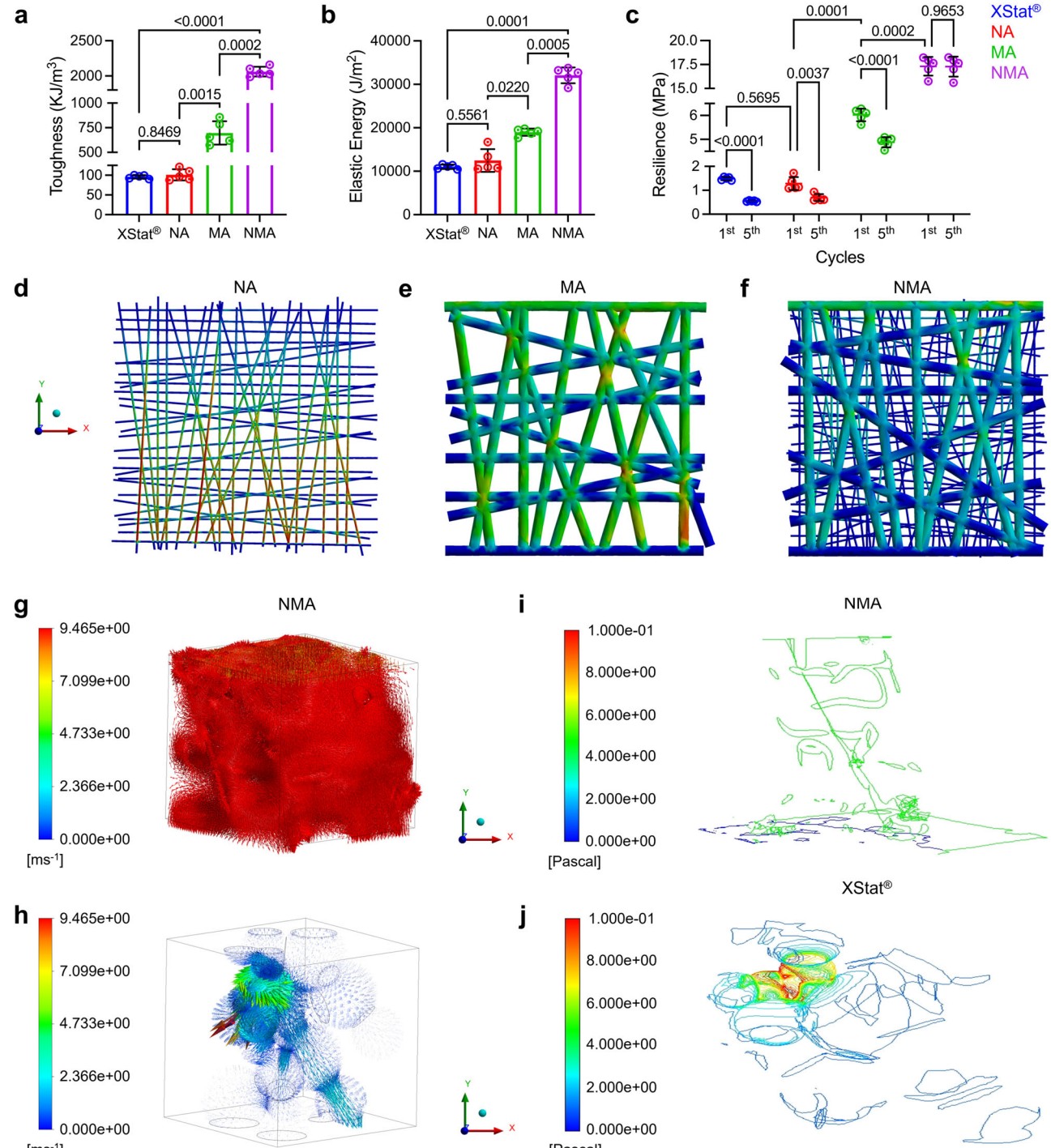

**Fig. 3 | Mechanical properties of samples, their simulated stress and blood flow distributions, and wall shear stress. a** Toughness. **b** Elastic energy under 90% strain. **c** Resilience from cycle 1 to cycle 5 under 90% strain. Data in (**a**–**c**) were shown as mean ± s.d. (*n* = 5 samples). Statistical analyses were conducted using one-way ANOVA (for **a** and **b**) or two-way ANOVA (for **c**), followed by Tukey's post hoc test. **d**–**f** The simulated stress distributions of NA, MA, and NMA under compressive forces. The distributions are normalized and color-coded using a rainbow scheme across the range of 0–1. **g**, **h** Fluid dynamics simulation of blood flow distributions in XStat® (**g**) and NMA (**h**). The blood flow distributions are normalized and color-coded using a rainbow scheme. **i**, **j** Fluid dynamics simulation of wall shear stress in XStat® (**i**) and NMA (**j**). The wall shear stress is normalized and color-coded using a rainbow scheme. NA: nanofiber aerogels, MA: microfiber aerogels, NMA: nano- and microfiber aerogels. Source data are provided as a Source data file.

## In vitro hemostatic efficacy

Before in vitro hemostatic efficacy testing, we investigated the hemocompatibility of the aerogels by measuring hemolysis. Supplementary Fig. 13a illustrates the color of the supernatants obtained via centrifugation. Notably, all aerogel groups exhibited a translucent color, while the control group displayed a distinct bright red color. Supplementary Fig. 13b showed that NA, MA, NMA, and XStat® had relatively low hemolysis ratios, indicating they are hemocompatible.

Next, we assessed the blood clotting capacity of the aerogels using a dynamic whole blood clotting assay. Figure 5a showed that the control and QuikClot® Combat Gauze (QCG®) groups had the highest

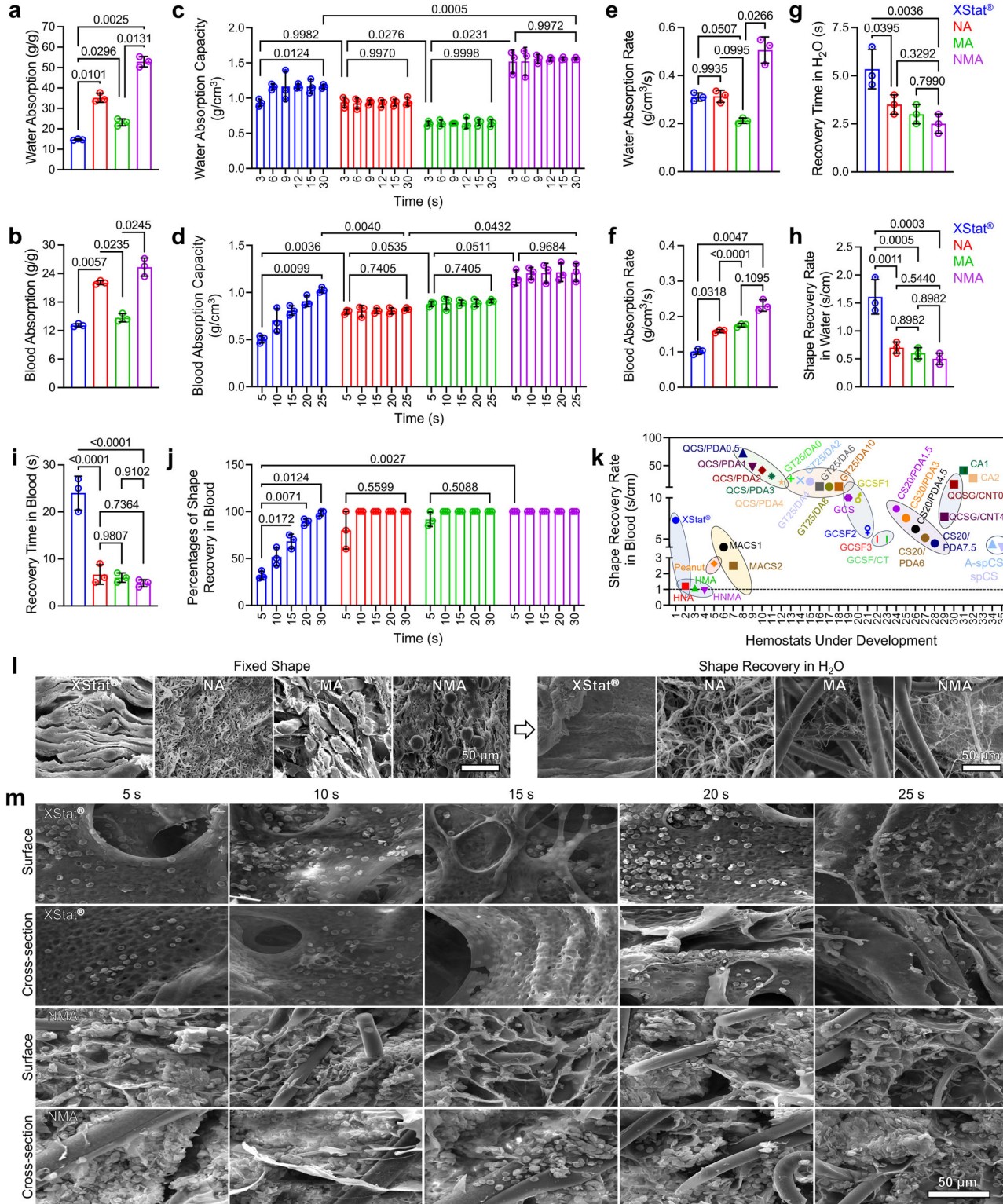

**Fig. 4 | Characterization of fluid uptake and shape recovery of samples.**
**a**, **b** water and human blood absorption kinetics of XStat®, NA, MA, and NMA. **c, d** Water (**c**) and human blood (**d**) absorption capacity at different times. **e**, **f** Water (**e**) and human blood (**f**) absorption rate. **g**, **h** Shape-recovery time (**g**) and rate (**h**) in water. **i, j** Shape-recovery time (**i**) and percentages of shape-recovery (**j**) at different times in human blood. Data were represented as mean ± s.d. (*n* = 3 samples). The significant difference was detected by one-way (for **a**, **b**, and **e**–**i**) or two-way (for **c**, **d**, and **j**) ANOVA with Tukey's multiple comparisons test. **k** Comparison of the shape recovery rate of NMA in blood with all materials reported in the literature known as

high mechanical resilient hemostats for junctional or non-compressible torso hemorrhage (NCTH). Values for comparison are given in Supplementary Table 2. **l** SEM images of samples after compression and shape recovery. **m** SEM images showing the surface and cross-sectional microstructure of NMA compared to XStat® after shape recovery in human blood at different time points. Micrographs in (**l**) and **m** are representative of 3 independent samples. NA: nanofiber aerogels, MA: microfiber aerogels, NMA: nano- and microfiber aerogels. Source data are provided as a Source data file.

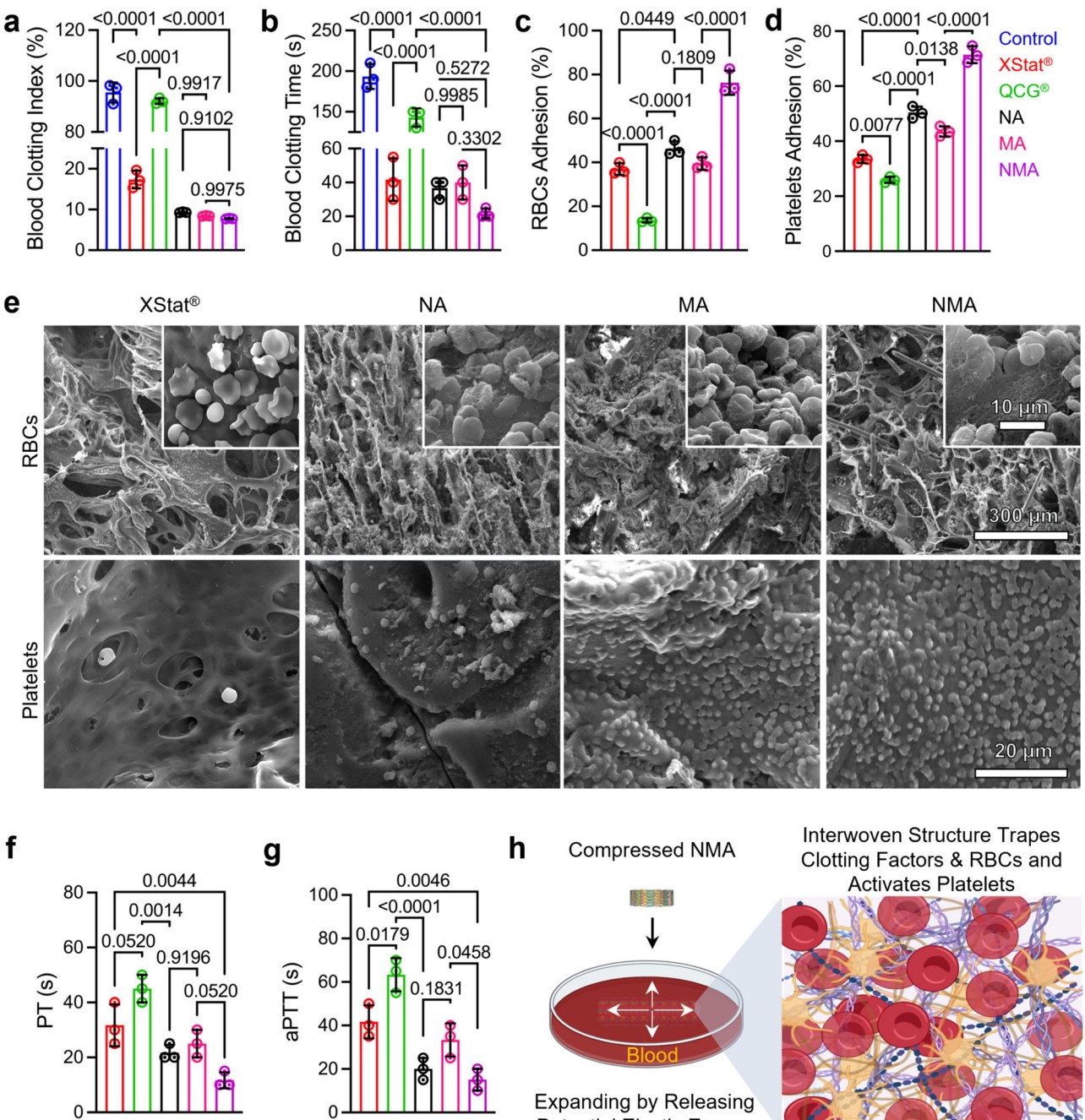

**Fig. 5 | Hemostatic efficacy in vitro. a** Blood-clotting index. The lower the index value, the higher the degree of blood clotting. **b** Whole blood clotting time. **c**, **d** Percentages of adhered RBCs (**c**) and platelets (**d**) on various samples. **e** SEM images depicting adhered RBCs and platelets in the cross sections of XStat® and aerogels. Micrographs are representative of 3 independent samples. **f**, **g** Prothrombin time (PT) (**f**) and activated partial thromboplastin time (aPTT) (**g**) for tested samples based on their capacity to initiate the activation of extrinsic and intrinsic pathways within the coagulation cascade. **h** Schematic diagram illustrating the pro-coagulant mechanism of NMA. Created in BioRender.

Shahriar, S. (2025) https://BioRender.com/y89s148. Data in (**a**–**d**), (**f**), and (**g**) were presented as mean ± s.d. (*n* = 3 samples). The significant difference was detected by one-way ANOVA with Tukey's multiple comparisons test. A separate *p*-value table for comparison of the percentages of platelets and RBCs adhesion between samples are provided as Supplementary Table 3. The significant differences in Supplementary Table 3 were detected by two-way ANOVA with Tukey's multiple comparisons test. QCG®: QuikClot® combat gauze, NA: nanofiber aerogels, MA: microfiber aerogels, NMA: nano- and microfiber aerogels. Source data are provided as a Source data file.

blood clotting index (BCI, which inversely correlates with the degree of blood coagulation), while the NA, MA, and NMA samples had lower BCI than XStat®. Figure 5b showed that the blood clotting times for the control, QCG®, and XStat® groups were 193.30 ± 15.27 s, 143.00 ± 11.26 s, and 41.67 ± 12.59 s, respectively. In contrast, the NA, MA, and NMA groups exhibited significantly shorter clotting times of 36.67 ± 5.78 s, 40.00 ± 10.00 s, and 21.67 ± 2.89 s, respectively.

We then evaluated the blood coagulation effects of various samples using RBC and platelet adhesion assays. The findings revealed that the adhesion percentages of RBCs (NA: 46.22 ± 3.32%; MA: 39.46 ± 2.84%) and platelets (NA: 50.41 ± 2.17%; MA: 43.50 ± 1.88%) for the NA and MA samples were notably higher compared to those for QCG® (RBCs: 13.63 ± 0.98%; platelets: 25.93 ± 1.12%) and XStat® (RBCs: 36.92 ± 2.65%; platelets: 33.49 ± 1.44%) (Fig. 5c, d and Supplementary

Table 3). Moreover, the NMA sample exhibited the highest adhesion percentages, with 76.31 ± 5.49% for RBCs and 71.44 ± 3.07% for platelets. SEM images corroborated these findings, showing fewer adhered RBCs and platelets in the cross-sections of XStat® and NA samples. In contrast, MA and NMA samples exhibited a high number of adhered RBCs and platelets (Fig. 5e). Additionally, platelets in the XStat® and NA samples maintained round morphology, whereas those in the MA and NMA samples appeared flattened and irregular in shape.

Finally, we assessed the impact of aerogels on the extrinsic and intrinsic coagulation pathways by measuring prothrombin time (PT) and activated partial thromboplastin time (aPTT) (Fig. 5f, g). The PT for the NMA group (11.67 ± 2.89 s) was significantly shorter than that of the XStat® (31.67 ± 7.63 s) and QCG® (45.00 ± 5.00 s) groups. Similarly, the aPTT for the NMA group (15.00 ± 5.00 s) was significantly lower than those of commercial products XStat® (41.66 ± 7.63 s) and QCG® (63.33 ± 7.63 s).

Based on the above findings, we speculated that the NMA technology has a pro-coagulant mechanism, as shown in Fig. 5h. The porous structure, high expansion ratio, and fast shape recovery of NMA allowed them to rapidly absorb a large volume of blood, exerting a potent blood-concentrating effect. These enhanced the coagulation process by promoting the aggregation and activation of RBCs and platelets and concentrating other plasma components. The nanofiber network within the aerogels also contained gelatin, which might further enhance platelet and blood cell adhesion and activation[29,31]. Together, these factors could contribute to the in vitro blood-clotting efficacy of NMA. By evaluating the physical characteristics and hemostatic properties, a weighted matrix was used to select the most optimized aerogel (Supplementary Tables 4, 5). Based on the weighted T-Score tables (Supplementary Tables 6, 7), NMA was chosen for further in vivo investigation.

## In vivo hemostatic efficacy

The experimental design adhered to the scientific rigor and transparency required by the National Institute of Health (NIH) and followed the ARRIVE guidelines (Supplementary Table 8)[32]. In the animals enrolled in the study, no significant differences were observed in body weight, baseline vitals (including MAP, heart rate [HR], end-tidal carbon dioxide [EtCO₂], and body temperatures), hematologic parameters (including hemoglobin, platelet count, and lactate levels), or coagulation profiles (PT, aPTT, international normalized ratio [INR], and fibrinogen levels) across the various groups (Supplementary Figs. 14–17). Figure 6a, Supplementary Figs. 18–23, and Supplementary Movies 12–16 showed the procedures of creating the porcine junctional hemorrhage model and the following treatment. The control group received no treatment after injury, highlighting the lethality of an untreated injury, while all treatment groups received their respective interventions after 30 seconds of free bleeding. The XStat® and QCG® groups served as primary and secondary clinical comparators, respectively, to the NMA. Figure 6b showed the application time of different treatment groups. Administering NMA took significantly less time, under 5 s, than QCG® (20.0 ± 3.7 s) and XStat® (8.4 ± 2.0 s). Notably, 100% of NMA-treated animals survived the 180-min study duration (Fig. 6c). In contrast, 20% of animals in the XStat® and 60% in the QCG® group succumbed within the first 60 min. Additionally, all untreated animals died from exsanguination between 10 and 45 min, losing 1603.2 ± 343.0 mL of blood after 30 s of free bleeding following the injury (Fig. 6c, d). Post-treatment blood loss in the NMA group (0.9 ± 0.2 mL/kg) was significantly lower than in the QCG® group (31.3 ± 13.7 mL/kg) and the XStat® group (17.5 ± 6.3 mL/kg) (Fig. 6d). The average blood loss, including the blood absorbed by the hemostats, was only 42.9 mL in the NMA-treated group, whereas the XStat® and QCG® groups had approximately 634.9 mL and 1265.3 mL, respectively (Fig. 6d).

Upon administration of the compressed NMA into the wound cavity, it rapidly absorbed blood and expanded, exerting pressure on the wound wall, and promptly achieving immediate hemostasis (Supplementary Fig. 24). In contrast, all other groups failed to achieve immediate hemostasis and exhibited continuous bleeding during and after application (Supplementary Movies 17–20). Initial hemostasis, defined as no bleeding after 3 min of manual pressure, was not observed in any subjects tested with QCG® and XStat® (Fig. 6e). Furthermore, there was no post-treatment rebleeding within 3 min of manual compression for the NMA group, whereas multiple incidences of rebleeding occurred in the QCG® and XStat® groups (Supplementary Movies 21–23). Consequently, stable hemostasis, defined as no bleeding throughout the study period, was exclusively achieved by NMA. Initial hemostasis was not achieved in any test animals in the QCG® and XStat® groups. For those that did not achieve immediate or initial hemostasis, delayed hemostasis was eventually attained at various times in nearly all surviving animals in the XStat® and QCG® groups. The mean time of hemostasis was depicted in Fig. 6f, showing that XStat® and QCG® required nearly 800 s to stop bleeding, while NMA uniquely achieved immediate hemostasis at the time (t) = 0 s in similar injury models (Supplementary Tables 9 and 10)[14,28,29,33–38].

Following severe injury, all animals treated with XStat® and QCG® were rebleeding, even after delayed hemostasis was achieved (Supplementary Movies 24, 25). Intriguingly, no instances of rebleeding were observed in animals treated with NMA (Supplementary Fig. 25). The rebleeding time for XStat® (795.0 ± 212.7 s) and QCG® (852.0 ± 407.9 s) was statistically non-significant compared to the continuous bleeding time in the control group (1428.0 ± 766.5 s), defined as the bleeding after 30 s of injury without treatment. In contrast, NMA showed no rebleeding (Fig. 6g). Notably, no clinically used hemostats or those under development have demonstrated a complete lack of rebleeding incidents (Supplementary Table 11)[1,14,33–36,38]. NMA exhibited no rebleeding in this porcine severe hemorrhage model, unlike QCG® and XStat®, marking a first in this context (Fig. 6h and Supplementary Movie 26). Interestingly, NMA was the only hemostat candidate in this severe injury model capable of achieving immediate hemostasis among all reported hemostats (Supplementary Tables 9–11). Rebleeding during compression and after delayed hemostasis in groups other than NMA contributed to a total increase in blood loss throughout the study period (Control: 2026.8 ± 336.4 mL, XStat®: 991.0 ± 245.3 mL, QCG®: 1671.0 ± 559.0 mL, and NMA: 554.1 ± 225.7 mL).

Following 30 s of free bleeding, the mean artery pressure (MAP) of swine in all groups decreased from baseline levels (Fig. 6i). After treatment, the MAP in the NMA group remained stable (47.00 ± 13.30 mmHg at 5 min and 47.20 ± 10.70 mmHg at 180 min), indicating immediate hemostasis. Conversely, MAP decreased to 24.40 ± 9.15 mmHg in the XStat® group, 11.40 ± 1.34 mmHg in the QCG® group, and 8.80 ± 3.11 mmHg in the control group within 5 min, significantly lower than the NMA group. This decline is likely attributable to delayed hemostasis and multiple rebleeding incidents. The decrease in MAP resulted in increased and unstable heart rate fluctuations in subjects treated with XStat® and QCG®, resembling those in the untreated group (Fig. 6j). Notably, the NMA group exhibited a consistent heart rate throughout the study, with no significant differences observed between 5 and 180 min (80.80 ± 10.50 beats/min at 5 min and 87.00 ± 21.52 beats/min at 180 min). Additionally, the NMA group demonstrated consistent hemoglobin, hematocrit, platelets, and lactate throughout the study period (Supplementary Table 12). In contrast, the hemoglobin level in the XStat® group decreased to 7.60 ± 0.90 from baseline within 15 minutes due to significant blood loss before achieving hemostasis (Supplementary Table 12). No discrepancies were observed in PT, aPTT, INR, and fibrinogen values in the NMA group (Supplementary Table 13). In contrast, the aPTT in the XStat® group increased by at least three times, contributing to

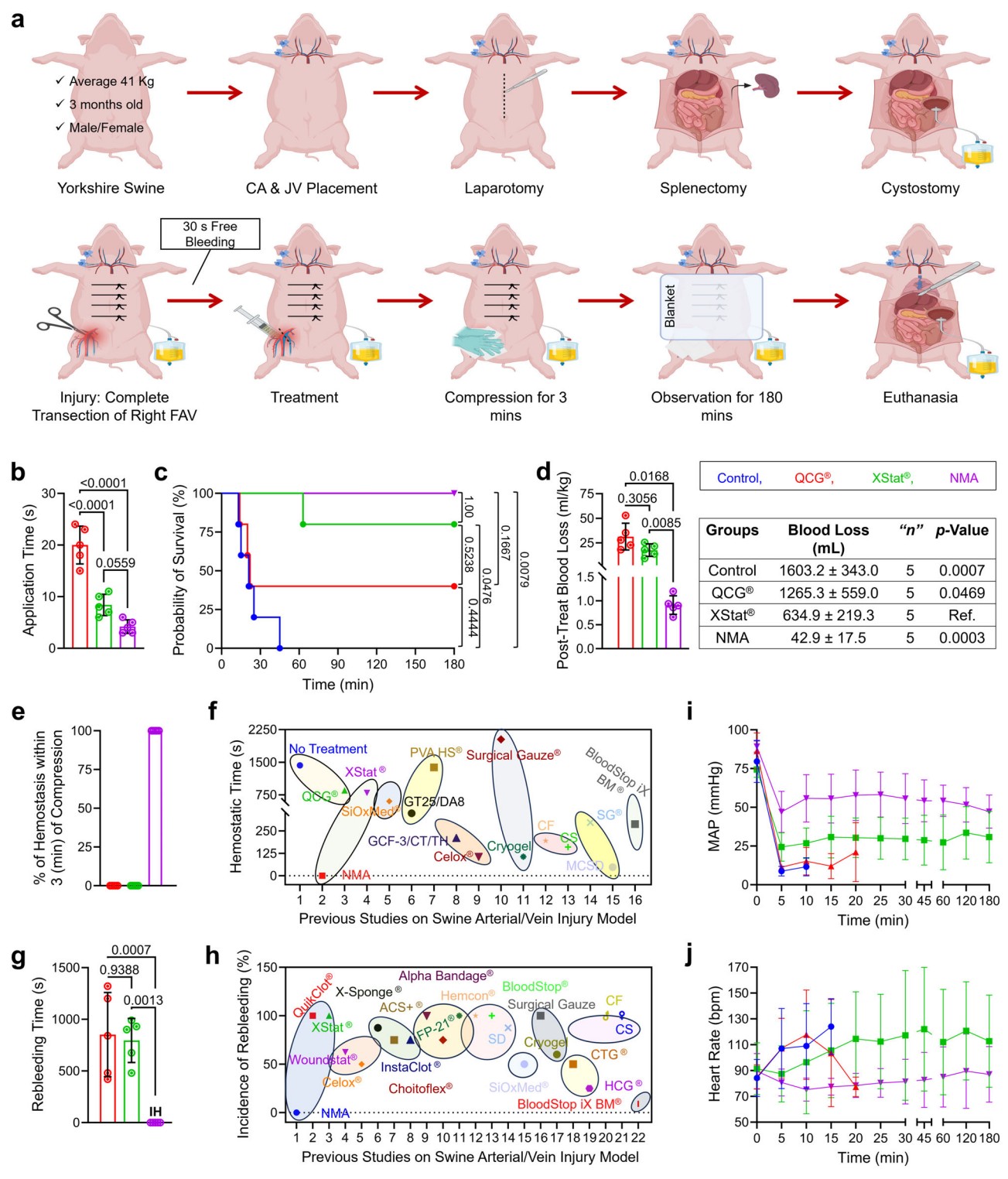

unstable hemostasis. Arterial blood gas parameters, including $HCO_3$, $PaO_2$, $PaCO_2$, and $EtCO_2$, remained stable in the NMA-treated group and the surviving subjects in the XStat® group (Supplementary Table 14). There was no significance in blood pH and body temperature between the XStat® and NMA groups (Supplementary Tables 15, 16). In addition, post-treatment observation of NMA within the wound cavity revealed that it effectively filled the wound area (Supplementary Fig. 26a). Upon removal, NMA conformed to the shape of the wound, indicating its ability to form a tamponade within the wound cavity without being expelled by excessive bleeding (Supplementary Fig. 26b). This characteristic suggests that NMA applies minimal pressure to the surrounding tissues, thereby reducing the potential risk of soft tissue injury. In contrast, XStat® showed inadequate expansion upon removal (Supplementary Fig. 26c, d). Furthermore, Supplementary Fig. 27 showed that NMA exhibited comparable tensile strength to XStat® after absorbing blood.

## Discussion

Uncontrolled bleeding is the major preventable cause of traumatic deaths globally[39], with more than half of these fatalities occurring before emergency care can be administered. In this study, we report a bicomponent aerogel for effectively managing a lethal JH in swine. The

**Fig. 6 | Hemostatic efficacy of NMA in a non-survivable swine femoral artery and vein complete dissection hemorrhage model, compared to QCG® and XStat®. a** Schematic depicting the creation of a lethal porcine junctional hemorrhage model and the process of treatment with injectable NMA. Created in BioRender. Shahriar, S. (2025) https://BioRender.com/y89s148. **b** Application time of NMA compared to QCG® and XStat®. **c** Kaplan–Meier survival plot. **d** Cumulative post-treatment blood loss, encompassing the wet hemostats utilized as treatment and the wet surgical gauzes applied for manual compression and covering the wound, in each group. **e** Percentages of subjects achieving hemostasis during 3-min compression. **f** Hemostatic time of NMA compared to those of previously reported materials that utilized a lethal swine artery/vein transection model. A summary of experimental conditions for comparison with this work and values from previously reported materials are provided in Supplementary Tables 9 and 10. **g** Time of continuous bleeding after treatment and manual compression in groups treated with QCG®, XStat®, or NMA. **h** Incidence of post-treatment rebleeding in the NMA

group compared with QCG®, XStat®, and all previously reported materials that utilized a similar type of animal model. Values from previously reported materials and a summary of experimental conditions for comparison with this work are provided in Supplementary Table 11. **i, j** Mean arterial pressure (MAP) and heart rate in subjects treated with various samples recorded over 180 min. Data were presented as mean ± s.d. in (**b**), (**d**), (**e**), (**g**), (**i**), and (**j**). $n = 5$ swine for all experiments. The significant differences in (**b**), (**d**), and (**g**) were determined by one-way ANOVA with Tukey's multiple comparisons test. The survival analysis in (**c**) used log-rank tests to compare survival curves across different groups ($n = 5$ swine). Statistical significance was assessed using the Fisher exact test (two-tailed) for categorical data comparisons. The significant difference for the table in (**d**) was determined by an unpaired t-test to compare values of XStat® with control, QCG®, or NMA. CA: carotid arterial catheter, JV: jugular venous catheter, FAV: femoral artery and veins, QCG®: QuikClot® combat gauze, **N**MA: nano- and microfiber aerogels, Ref: Reference, IH: immediate hemostasis. Source data are provided as a Source data file.

aerogel mainly consisted of PLA nanofibers and PCL microfibers, which entangle, forming a highly porous structure with a high percentage of open pores. In designing the aerogels, several key features were taken into account: (i) the ability to recover their shapes upon deployment and endure internal pressures within the wound environment on the bleeding site; (ii) resilience to high compressive forces in their dry state, ensuring the aerogel's integrity as an injectable hemostat; (iii) open pores larger than blood components, particularly RBCs (6–8 μm), to ensure efficient blood absorption; and (iv) the capacity to trap clotting factors and blood components, such as RBCs and platelets, within the aerogel to facilitate the coagulation cascade and prevent post-treatment bleeding. The NMAs were synthesized and evaluated as an injectable hemostat following the above-delineated design criteria. It is noteworthy to mention that sample preparation can be optimized through the use of molds specifically designed to produce multiple samples simultaneously. Alternatively, large aerogel samples could be produced and subsequently cut into smaller sizes as needed. To ensure the structural integrity of the material and prevent damage to the edges during cutting, frozen samples would be utilized. Another approach would be the preparation of large-sized aerogel materials that could be compressed and packaged within a dissolvable net bag. This technique would allow for the direct application of the compressed material into a wound cavity. Upon administration, the net bag would dissolve immediately, enabling the material to expand to its original shape and effectively fill the wound site. This method offers a practical advantage by eliminating the need for small pellets, streamlining the preparation process, and enhancing ease of use in clinical and field settings.

Materials used in junctional wounds for hemostasis must have strong elastic properties to re-expand and fill the cavities, creating a tamponade effect[19]. The only FDA-approved shape expandable hemostat, XStat®, and many other products under development for managing JH and non-compressible torso hemorrhage (NCTH) might exhibit suboptimal mechanical properties and structures, which could potentially limit their effectiveness in hemostasis. The biocomponent aerogel demonstrated tunable mechanical properties, including a high specific elastic modulus, high resilience, rapid shape recovery, and exceptional robustness. These attributes were likely due to its intricate network composed of dual-sized fibers, along with gelatin coating and cross-linking. Optimizing the elastic energy release from shape-expandable hemostatic pellets is crucial. Insufficient compression modulus could result in a slow expansion rate, worsening blood loss from the wound, while excessive compression modulus could exert severe pressure, leading to additional soft tissue injury during application. In this study, the elastic modulus measurements under different compression rates demonstrated that NMA could recover its shape effectively even after 90% compression. Interestingly, the compression modulus of NMA (ranging from 7.87 ± 2.81 kPa to 961.85 ± 48.69 kPa) was found to be lower than those of human soft tissues (ranging from

several kPa to several MPa)[40]. Notably, the mechanical strength of the aerogels could be finely tuned by adjusting cross-linking time without compromising their resilience. This highlights the versatility of these materials in achieving the desired balance between compression strength and elasticity. Additionally, the 100-cycle compression-relaxation test demonstrated fatigue resistance and mechanical stability. These properties are crucial for maintaining a strong and stable hemostatic barrier, which applies consistent pressure on surrounding tissues to effectively control bleeding.

Simulated results of stress distributions under compressive forces demonstrated that NMA could exhibit the lowest level of stress and the most uniform stress distribution among the aerogels, indicating their superior capability in elastic energy transfer and storage during compression. Fluid dynamic simulations of blood flow distributions and wall shear stress further revealed that NMA could absorb blood faster and had a higher blood absorption capacity than XStat®. Experimental results confirmed that NMA had the highest blood absorption rate among the tested samples. Unlike expandable hemostats generated by freeze casting or gas foaming methods, which exhibited low interconnectivity and hindered blood penetration, NMA demonstrated a faster shape recovery rate in blood than XStat® and other reported hemostats[4,13,14,16,17,19,28–30]. Previous studies highlighted the significant role of pores in directing blood flow into hemostatic materials and facilitating the accumulation of clotting components[4,10–12,14,16–18,28,29]. Due to its large percentage of open pores and trape nest-like structures, NMA accumulated and adhered significantly more RBCs and platelets than XStat®. Consequently, NMA achieved a shorter blood clotting time than XStat®. Additionally, NMA exhibited lower PT and aPTT values than XStat®, which suggested that these aerogels activated both the extrinsic and intrinsic coagulation pathways.

NMA demonstrated a shorter application time compared to QCG® and XStat® for material administration. Notably, all five animals in the NMA group survived throughout the 180 min of study, a result not matched by other tested materials in similar studies[13,14,28,29,33–35]. This group also experienced the most minor blood loss. NMA required the shortest time to achieve hemostasis among all materials tested in this study and those reported in the literature[14,28,29,36–38]. Additionally, NMA showed no incidence of rebleeding. This injectable aerogel could be beneficial for treating NCTH as well. We propose the hemostatic mechanism of NMA in JH as follows: (i) compressed dry NMA store elastic energy due to efficient stress transmission by the bi-network; (ii) upon injection into a bleeding cavity, the porous fiber architecture rapidly absorbs blood, allowing the aerogel to return to its original shape; (iii) the shape recovery effect exerts pressure on the wound, creating a physical barrier that quickly stops bleeding; (iv) the intertwined fibrous structure concentrates coagulation factors, enhancing the aggregation of erythrocytes and activated platelets; and (v) RBCs and platelets accumulate on the surface and within aerogels and are activated, achieving stable hemostasis and preventing rebleeding. In

contrast, the hemostatic mechanisms of XStat® and QCG® differ significantly from NMA. XStat® employs chitosan granules that promote platelet adhesion and aggregation while interacting with red blood cells and proteins, initiating clot formation. QCG®, on the other hand, relies on kaolin, an inorganic mineral that activates Factor XII in the coagulation cascade to generate thrombin and form fibrin clots[41]. However, these mechanisms are limited in addressing the complexity of severe JH or NCTH, as evidenced by their reduced effectiveness in our non-survival uncontrolled JH model.

Previous studies demonstrated cell infiltration, new blood vessel formation, and ECM deposition within hybrid aerogels after subcutaneous implantation in rats[22]. Beyond its potential application in JH, NMA could also be used for wound healing and tissue regeneration due to its porous biomimetic architecture. Additionally, the super elastic property and shape recovery property of NMA enable them to combine with cells to form 3D tissue constructs that could be implanted in a minimally invasive manner[22,42–45]. Biological molecules could be incorporated into NMA through various methods (e.g., coating, conjugation, and encapsulation) to regulate cell behavior or infection management, further enhancing their capability in tissue regeneration[46,47]. While the current NMA formulation lacks antibacterial properties, incorporating antimicrobial agents, such as chitosan or antimicrobial peptides, could prevent infection at wound sites. However, this study focuses on the prehospital management of lethal JH where NMA is intended for short-term use to stabilize patients during transport to the hospital. Given concerns about increased production costs, manufacturing complexity, and regulatory hurdles, antibacterial agents were not included at this stage. Future studies may explore antimicrobial incorporation for extended applications, such as infectious wound treatment, where NMA's biodegradable nature could offer additional therapeutic benefits. In contrast to non-biodegradable XStat®, which requires enclosed pellets for their complete removal, NMA's biodegradability allows portions of the material to remain in the wound, potentially aiding tissue regeneration post-application[22].

A limitation of this study is that we had to reuse recently published survival and post-treatment blood loss data for the control, XStat®, and QCG® groups for comparison purposes, as they were obtained from the same protocol. This approach was chosen to align with the 3Rs principle of animal research, reducing the number of animals used while ensuring robust control data, as the data were collected under the same experimental protocol. Additionally, the study faced challenges in blinding the investigators to the treatment types during the porcine procedures, although the UNMC Comparative Medicine's administrative officer who recorded all vitals and the pathologist who collected all pathological data were blinded. Other limitations include the use of castrated males instead of intact males and conducting the porcine model under general anesthesia, which may not fully replicate the physiology of an awake patient. The necessity of performing a splenectomy in swine hemorrhage models remains a topic of debate[48,49]. Furthermore, the requirement for splenectomy is highly model-dependent, influenced by factors such as the rate and volume of blood loss, the hemorrhage endpoint, and the specific study objectives[48]. While splenectomy may not be essential in controlled hemorrhage models with moderate blood loss rates and fixed-pressure endpoints, its role in uncontrolled hemorrhage models like ours, characterized by rapid and severe blood loss, is unclear. Further studies are needed to determine whether splenectomy is necessary in such models and to evaluate its impact on hemostasis, inflammation, and overall physiological responses. Also, we did not assess the material's stability under extreme environmental conditions, such as high temperatures, humidity, or pressure, which are critical for battlefield or emergency applications. Future studies are necessary to address these conditions and validate the aerogels' functionality in such scenarios.

## Methods

### Fabrication of biocomponent aerogels

The bicomponent aerogels were fabricated through a series of processes, including electrospinning, wet electrospinning, freeze-casting, and cross-linking[22]. The initial step involved creating short NFs from an NF mat. Specifically, 2 g of PLA (MW: 60 kDa) pellets (Sigma-Aldrich, St. Louis, MO, USA) were dissolved in a 20 mL solution consisting of 16 mL dichloromethane (DCM) (Oakwood Chemical, Estill, SC, USA) and 4 mL dimethylformamide (DMF) (Oakwood Chemical). To enhance hydrophilicity, 0.01 g of Pluronic F-127 (Sigma-Aldrich) was added to the PLA solution. This solution was then loaded into a 20 mL syringe for electrospinning. The electrospinning process was carried out at a flow rate of 0.6–0.9 mL/h using a syringe pump (Fisher Scientific, Pittsburgh, PA, USA) under a voltage of 15–18 kV with a 21-gauge needle (the spinneret) connected to a high voltage generator (ES304-5W/DAM, Gamma High Voltage Research Inc, Ormond Beach, FL, USA). Nanofibers were collected on a grounded mandrel, which was 10 cm long, 12 cm in diameter, and rotating at 1200 rpm, positioned 13 cm from the needle. After collection, the NF mat was removed from the collector, air-dried for 24 h, and then treated with air plasma (PDC-001-HP, 115 V, Harrick Plasma, New York, USA). The fibers were cut into lengths of 60 μm, 30 μm, and 20 μm using a cryostat, then dispersed in water to create suspensions, freeze-dried, and stored at 4 °C for further use.

The subsequent step involved the fabrication of short MFs using wet spinning. To prepare a 20% w/v PCL (MW: 80 kDa) (Sigma-Aldrich) solution, 2 g PCL pellets were dissolved in 10 mL mixture of DCM and DMF, with each solvent contributing 5 mL. This solution was then loaded into a 20 mL syringe and extruded at a rate of 3.0 mL/h through a 3D printed device equipped four 21-gauge needles. The extruded solution was directed into a coagulation bath containing 200 proof ethanol (Decon Lab, King of Prussia, PA, USA). The extrusion device, designed with four outlets and one inlet, was created using a Vida 3D printer (EnvisionTEC, Gladbeck, Germany) and fabricated from Clear Guide material (EnvisionTEC). A low-speed plastic mandrel wrapped in aluminum foil, positioned above the ethanol bath, was used to collect the solidified MFs. After the wet spinning process, the MF bundle was thoroughly dried for several days, cryostat cut under liquid nitrogen to prevent pressure-induced fusion, and treated with air plasma using a High Power Expanded Plasma Cleaner (PDC-001-HP model, 115 V, Harrick Plasma, New York, USA).

Next, short NFs, MFs, or at 50/50 w/w combination of both (NF/MF) were dispersed in water to form suspensions. These dispersed fibers were then homogenized to prepare using a probe homogenizer set at 20% amplitude, with 20/10 s on/off cycles, for 2.5 h under ice-cold conditions. Following homogenizations, the short NF, MF, or NF/MF suspensions were mixed with 1% gelatin granules (derived from bovine skin, Type B, Sigma-Aldrich), and further homogenized for another 2.5 h under the same conditions. The resulting homogenized suspension was then poured into a copper mold attached to an aluminum plate and quickly transferred to a −80 °C freezer, where it was left for 12 h. After freeze-drying, the samples were cross-linked by exposure to glutaraldehyde vapor (EM grade, 2.5% in anhydrous ethanol, Ladd Research, Cincinnati, OH, USA) for 6, 12, or 18 h. NA and MA were created using the same method as NMA. The key difference is that NA lacks MFs, while MA lacks NFs. Finally, the samples were sterilized using ethylene oxide gas (Anprolene AN7916 Ethylene Oxide Ampoules, Andersen Sterilizers Inc., Haw River, NC, USA).

### Physical characterization and analysis

The aerogel samples were weighed using a digital balance (U.S. Solid-Analytical Balance, Cleveland, OH 44103, USA). The balance was calibrated before each session to ensure accuracy. Measurements were repeated three times for each sample, and the averaged values with standard deviations were recorded to ensure precision.

To capture the intricate details of the samples, images were taken using a Galaxy Note 20 Ultra (Samsung, Pyeongtaek, Gyeonggi, South Korea). A scale was included in the field of view during imaging, allowing for subsequent calibration and dimensional analysis of the aerogels. The diameter and length of aerogels were measured using Fiji software by analyzing these microphotographs[50].

For a comprehensive assessment of the internal structure and morphology of the aerogels, micro-computed tomography (micro-CT) analysis was performed. The cylindrical aerogel samples were securely mounted in a specialized holder using a single-sided tape to ensure stability during scanning. A Bruker SkyScan 1276 - CMOS Edition micro-CT scanner (Kartuizersweg, Kontich, Belgium) was used, with scanning parameters meticulously configured to optimize resolution and imaging quality. Initial surface area and porosity analyses were conducted to examine the structural characteristics of the aerogel. Cross-sectional and longitudinal images were then captured through a rotation mechanism within the micro-CT scanner, with images taken from various angles to build a comprehensive representation of the internal structure. The micro-CT data was reconstructed using CTVox SkyScan software, which employed advanced algorithms to convert raw projection data into high-resolution 3D images. The visual analysis was further enhanced by generating 3D reconstructions and virtual slices, followed by in-depth analysis utilized CTVox software tools for quantitative metrices like voxel-based density distribution and porosity mapping, including open pores and close pores.

The cross-sectional morphology of the aerogel samples was further investigated using SEM (FEI Quanta 200, Hillsboro, OR, USA). The aerogel specimens were first attached to SEM stubs using a conductive adhesive for grounding, then sputter-coated with a thin layer of conductive material, such as gold, to augment surface conductivity and mitigate charging effects during imaging. This coating procedure was performed in an argon gas environment at a voltage of 110 milliamperes. The coated samples were placed in the SEM chamber under high vacuum conditions, with nitrogen gas present. Before imaging, the SEM parameters including acceleration voltage (25.0 kV), working distance, and aperture size, spot (3.0), and dwell time (1–3 μs)—were carefully adjusted to optimize imaging conditions for the aerogel's cross-sectional morphology. SEM images were taken at various magnifications to capture the fine details of the internal structure, focusing on cross-sectional views to reveal the pore composition and distribution within the material.

Pore diameter measurements were conducted using ImageJ software, following the acquisition of clear, high-resolution SEM images of the aerogel samples. The images were calibrated using the scale bar where applicable, converting pixel measurements to real-world units. Image processing techniques, such as contrast and brightness adjustments, were applied to enhance pore visibility. Pore diameters were measured using Fiji's 'Straight Line' tool, with a line drawn across a representative pore through its center. For calibrated images, the software provided measurements in real-world units; for non-calibrated images, known scale information was used to convert the measurements. Multiple measurements were taken across different pores and regions of the SEM image to ensure representativeness. The data analysis in Fiji generated a spreadsheet of the recorded measurements, from which the average pore diameter and standard deviation were calculated when multiple measurements were recorded. Likewise, the diameter and length of short NFs and MFs were quantified using Fiji analysis of their SEM images. The volume of the cylindrical aerogels or XStat® was determined using the Eq. (1).

$$V = \pi \times r^2 \times h \qquad (1)$$

In Eq. (1), "V" represents the volume of the cylindrical structures, "r" denotes the radius, and "h" signifies the height.

## Mechanical properties

A compression test was conducted to assess the mechanical properties of the aerogel samples, which were cylindrical in shape. The testing was performed utilizing an Instron 5640 universal test machine (CellScale Biomaterials Testing, Waterloo, Ontario, Canada). The samples were secured within petri dishes using double-sided tape and then mounted onto the lower compression plate of a CellScale Univert apparatus (Serial Number: UV55290, CellScale Biomaterials Testing, Waterloo, Ontario, Canada). To simulate physiological conditions, the petri dishes were filled with citrated whole human blood (courtesy of the UNMC Blood Bank, Omaha, NE, USA). The compression test began by applying a load of 200N to induce compressive deformation at displacement levels of 20%, 40%, 60%, 80%, and 90%, all at a controlled rate of 1 mm/s. After each compression cycle, the samples were held for 5 min to allow for stress relaxation within the aerogel structure. To assess the shape recovery capabilities of the aerogel specimens, the compressive force was completely released after the 5-min holding period. The aerogels were then submerged in human blood following each compression levels of 20%, 40%, 60%, 80%, and 90%. The lengths of the aerogels were carefully measured before and after recovery using a digital caliper with a resolution of 0.01 mm. The study determined the maximum compressive strength at various deformation levels (20%, 40%, 60%, 80%, and 90%) after each compression cycle. To ensure consistency and reliability, three samples from the same batch were tested in each experiment. The specific elastic modulus was calculated using the following Eq. (2).

$$\text{Specific Elastic Modulus} = E/m/V \qquad (2)$$

In Eq. (2), "E" represents the elastic modulus of the tested samples and "m" and "V" denote mass and volume, respectively. The volume of the cylindrical aerogels was determined using Eq. (1). The elastic modulus (E) was calculated using the following Eq. (3)[51].

$$E = (F \times H_0)/(A \times \Delta H) \qquad (3)$$

In Eq. (3), "F" represents the compressive force, "A" is the cross-sectional area of the aerogels, $H_0$ denotes the initial height, and ΔH indicates the change in length after compression.

We then aimed to investigate the impact of cyclic compressive strains on the mechanical properties, including strength and flexibility, of samples. The experiment involved subjecting each sample to cyclic compressive strains of 70%, 80%, and 90% over five rounds. Each round consisted of three repetitive cycles at the designated compressive strains, with each cycle involving 10 s of compression, 5 s of holding, 10 s of recovery, and 5 s of rest. The sample was first subjected to 70% compression, followed by 80% and finally 90% in subsequent rounds. To assess comparable metrics of compression resistance among the aerogels, we calculated critical forces, max forces, and force loss. Critical force, representing the force required to initiate compression, was determined as the maximum force observed at the peak of the force-displacement curve. Max force was defined as the maximum compressive force achieved during the specified displacement levels (70%, 80%, and 90%). Force loss was calculated as the percentage difference between max forces in consecutive cyclic compressions. Moreover, the change in length of the aerogels after cyclic compression tests was evaluated as the percentage difference between maximum changes in length across successive cycles. For comparison, different samples were subjected to cyclic compression. XStat®, NA, and MA samples underwent 5 cycles of cyclic compression at 70%, 80%, and 90% displacement, while NMA samples underwent 50 cycles of cyclic compression at 90% compressive strain to assess mechanical robustness. The compression, hold, recovery, and rest durations were standardized at 5, 1, 5, and 1 s per cycle, respectively, with a compression and relaxation rate of 1 mm/s.

To further investigate the influence of cross-linking time on the mechanical properties of NMA, cyclic compression-relaxation tests were conducted to assess their compressive strength and changes in length over multiple cycles. Two sets of NMA were prepared with different cross-linking durations: one set cross-linked for 6 h and the other for 18 h. Initially, both sets underwent cyclic compression tests for 10 cycles at 90% strain per cycle, with standardized duration for compression, hold, recovery, and rest at 5, 1, 5, and 1 s, respectively, and a compression and relaxation rate of 1 mm/s. Force measurements were recorded at 90% strain during compression. Subsequently, the aerogels with a 6-h cross-linking time were subjected to an extended test of 100 cycles to evaluate their mechanical robustness, maintaining the same compression protocol. The change in length after cyclic compression was assessed by measuring the percentage difference between the maximum changes in height observed across consecutive cycles.

To assess the toughness of XStat®, NA, MA, and NMA samples compression tests were conducted at a displacement level of 90%. Each test cycle included standardized durations for compression, hold, recovery, and rest, which were set at 10, 60, 10, and 60 s, respectively. The compression and relaxation rates were maintained at 1 mm/s throughout the test. Toughness was determined by analyzing the stress-strain curve and calculating the area under the curve using the Eq. (4).

$$T = \int_0^{\epsilon f} \sigma . d\epsilon \qquad (4)$$

Where "T" is the toughness, "$\sigma$" represents stress, "$\epsilon$" denotes strain, and "$\epsilon f$" indicates strain at failure. We then conducted five cyclic compression tests to measure the hemostats' elastic energy absorption and resilience. Each test involved compressing the samples to a strain of 90% at a speed of 1 mm/s. The compression cycle included four phases: compression, hold, relaxation, and rest, each lasting 10, 5, 10, and 5 s, respectively. To quantify the energy absorbed during compression at 90% strain, we recorded the maximum compressive force applied to the hemostats and the corresponding displacement. These measurements allowed us to calculate the stress and strain at various points during the compression process. The area under the stress-strain curves was then computed to determine the energy absorbed by the aerogels during compression.

Resilience (denoted as R) was calculated using Eq. (5) by determining the area under the stress-strain curve up to the elastic limit. This provides insight into the aerogel's ability to store and release energy under loading and unloading.

$$R = \frac{1}{2} \times \sigma y \times \epsilon y \qquad (5)$$

Where "$\sigma y$" and "$\epsilon y$" represent yield strength and yield strain.

**Simulations**

The mechanical properties of NA, MA, and NMA were evaluated using finite element method (FEM) simulations conducted with the Student Version of ANSYS 2024 R1. Compressive forces were applied along the Y-axis, and the resulting distributions of Von Mises stress were analyzed. To understand how the different architectures influence stress transmission, the applied force was normalized by the mass of each sample.

Additionally, FEM simulations for NMA and XStat® were conducted using fluid flow analysis in ANSYS Student 2024 R1. Blood considered an incompressible fluid in these simulations, was assigned a density of 1060 kg.m$^{-3}$ with a zero rate of density change. The governing equations for these flow simulations included the continuity equation and the 3D incompressible Navier-Stokes equation, with the continuity equation expressed as Eq. (6).

$$\frac{\partial \rho}{\partial t} + \nabla \cdot (\rho \vec{v}) = 0 \qquad (6)$$

Where $p$ is density, $v$ is velocity, and $\nabla$ is gradient operator. The gradient operator was computed by the following Eq. (7).

$$\vec{\nabla} = \vec{i} \frac{\partial}{\partial x} + \vec{j} \frac{\partial}{\partial y} + \vec{k} \frac{\partial}{\partial z} \qquad (7)$$

In addition, the Navier-Stokes equation can be expressed as Eq. (8).

$$\rho \left( \frac{d\vec{v}}{dt} + \vec{v} \cdot \nabla \vec{v} \right) = -\nabla p + \mu \nabla^2 \vec{v} + f \qquad (8)$$

In Eq. (8), $t$ is time, $p$ is the fluid pressure, $\mu$ is the fluid dynamic viscosity, and $f$ is the external forces applied to the fluid.

For the viscosity $\mu$, the non-Newtonian behavior of blood flow is modeled using the Carreau model, which is given below as Eq. (9).

$$\mu = \mu_\infty + (\mu_0 - \mu_\infty)(1 + (\lambda \dot{\gamma})^2)^{\frac{n-1}{2}} \qquad (9)$$

In Eq. (9), $\mu_0$ and $\mu_\infty$ are the zero and infinite shear rate viscosities, respectively, and $\lambda$ is the relaxation time constant. For the case of blood, $\mu_0 = 0.056$ kg m$^{-1}$ s$^{-1}$, $\mu_\infty = 0.0035$ kg m$^{-1}$ s$^{-1}$, $\lambda = 3.313$ s, $n = 0.3568$.

In all numerical simulation projects, a steady-state approach was used for the calculation. The blood flow velocity in the inferior was set at 10 m/s, with the flow assumed to be laminar. The outlet was designated as the outflow. The calculation parameters were as follows: a time step size of 0.01, 50 time steps, and a maximum 200 iterations per time step.

**Fluids uptake behaviors**

A series of experiments were conducted to assess the fluid absorption properties of NA, MA, and NMA in comparison to XStat®. Initially, the dry and compressed forms of each sample were exposed to either water or human blood (UNMC Blood Bank). The physical appearance of the samples immersed in blood was thoroughly documented using a digital camera (Apple 14 Pro Max, Cupertino, CA, USA), with the images presented unaltered. Additionally, the pH values of the diluted blood after absorption were measured using a pH meter (Orion Star™ A221 Portable pH Meter, Thermo Scientific, Waltham, MA, USA).

For quantitative analysis, the compressed forms of NA, MA, NMA, or XStat® were weighed initially (denoted as "W$_d$" in g). These materials were then immersed in either water or human blood. After a precise interval of 30 s, the materials were removed and re-weighed (denoted as "W$_W$" in g). Given the variability in the dry weight of the aerogels (e.g., NA, MA, or NMA) and XStat®, the following Eq. (10) was employed to calculate the fluid absorption of the materials in terms of weight/weight.

$$\text{Fluid Absorption (g/g)} = (W_W - W_D)/W_D \qquad (10)$$

To further evaluate the water and blood absorption capacities of the aerogels (e.g., NA, MA, and NMA) in comparison to XStat®, the study was extended to include volume variations. Initially, the volume of each material (NA, MA, NMA, or XStat®) was measured using Eq. (1), and recorded as V (cm$^3$). The dry and compressed samples were then immersed in either blood or water. Their positions in the liquids were meticulously recorded using a digital camera (Apple 14 Pro Max). The samples were removed from the liquids at specific intervals $-3$, 6, 9, 12, 15, and 30 s for water, and 5, 10, 15, 20, and 25 s for blood. The water or

blood absorption capacity was subsequently calculated using Eq. (11).

$$\text{Fluid Absorption Capacity}(g/cm^3) = (W_W - W_D)/V \tag{11}$$

The water and blood absorption rate, expressed in $g/cm^3/s$, were determined by calculating the slope of the water or blood absorption capacity versus time curve. This calculation was performed using Eq. (12).

$$\text{Fluid Absorption rate}(g/cm^3/s) = (W_W - W_D)/V/\text{time} \tag{12}$$

## Shape memory properties

A series of experiments were conducted to assess the shape-memory properties of NA, MA, and NMA compared to XStat® in both water and blood. The aerogels were initially sized at at $5.5 \times 1\,cm^2$, while XStat® was sized at $3.3 \times 1\,cm^2$. The aerogels underwent compression to form pellets. Subsequently, the NA, MA, and NMA pellets were exposed to either water or blood. The entire shape recovery process, along with the associated timeframes, was meticulously recorded using a digital camera (Apple 14 Pro Max). The volume of the samples before and after shape recovery was measured using Eq. (1). The shape recovery time, defined as the duration required for the aerogels to fully return to their original shape, was measured with a stopwatch (Fisherbrand™ Traceable™ Four-Channel Countdown Stopwatch with Memory Recall, Fisher Scientific). Given the variations in length between NA, MA, NMA ($5.5 \times 1\,cm^2$) and XStat® ($3.3 \times 1\,cm^2$), the shape memory time was normalized and calculated in s per cm using Eq. (13). This provided a standardized measure of the time each sample required to recover its shape per centimeter when immersed in water or blood.

$$\text{Shape Memory Rate (s/cm)} = \text{Total Time to Shape Recovery}/ \\ \text{Total Length of Materials} \tag{13}$$

A comparative analysis was performed to assess the shape recovery rate of dried NMA pellets in blood, comparing it with data on wet and dry materials reported in the literature over the past 20 years. This analysis involved a comprehensive search of PubMed, Scopus, and Google Scholar databases, focusing on materials known for their high mechanical resilience as hemostats for JH or NCTH. Keywords used in the search 'shape memory materials', 'aerogels', 'hemostats', 'cryogels', 'sponges', 'foam', 'JH, 'rapid hemorrhage', 'hemostasis', and 'NCTH'. The information gathered provided valuable insights into the shape recovery characteristics of NMA relative to established materials in the field.

In addition, experiments were conducted to evaluate the shape recovery percentages of NA, MA, and NMA in comparison to XStat® at various time points in blood. Initially, the aerogels were compressed to fix their shapes and then exposed to water and blood. The shape recovery percentages were recorded in water after 5 s of contact. For blood, samples were extracted at intervals of 5, 10, 15, 20, and 30 s to assess the shape recovery over time. The percentages of shape recovery in both water and blood were calculated using Eq. (14).

$$\text{Shape Recovery}(\%) = L_R/L_T \times 100 \tag{14}$$

In Eq. (14), "$L_R$" is the recovered length at specific time points, and "$L_T$" is the total length of the sample. The shape recovery ratios in water and blood were documented using a digital camera (Apple 14 Pro Max).

The reversible absorption properties of water-soaked NMA were quantified using a manual compression method. The aerogels, with a diameter of 10 mm and a height of 7 mm, were compressed axially from the top. This compression process caused the aerogels to release water, which was then reabsorbed once the force was removed.

The amount of water released, and the subsequent reabsorption were meticulously recorded using a digital camera (the Samsung Galaxy Note 20 Ultra 5G).

Additionally, the microstructure recovery of NA, MA, NMA, and XStat® before and after absorbing water was analyzed using SEM. This analysis also examined the relationship between shape recovery in human blood and the absorption of blood cells on the surface and cross-sectional areas at various time points. To prepare the samples for SEM analysis after blood absorption, the chemical-dry method was employed. Samples were first rinsed with 1× Dulbecco's Phosphate Buffered Saline (DPBS) (Thermo Fisher Scientific, Waltham, MA, USA) and then fixed in a solution of 2% paraformaldehyde (PolySciences, Warrington, PA, USA) and 2.5% glutaraldehyde (Ladd Research, Williston, VT, USA) in 0.1 M Sorenson phosphate (Electron Microscope Sciences, Hatfield, PA, USA) for 3 h at room temperature. Following fixation, the samples were thoroughly washed with DPBS to remove excess fixative. The samples were then treated with 1% osmium tetroxide (Sigma-Aldrich) for 30 min at room temperature, followed by three additional washes with DPBS. For dehydration, a graded ethanol series (Fisher Scientific) was used, progressing through 35%, 50%, 70%, 95%, and 100% ethanol, with each concentration applied for 5 min. The samples were further treated with a graded series of hexamethyldisilazane (HMDS) (Electron Microscope Science) at 30%, 70%, and 100% concentrations, each for 5 min. The final step involved allowing the samples to air-dry in 100% HMDS within a chemical hood.

## Pro-coagulant hemostatic efficacy evaluation

To test the hemostatic efficacy of the tested samples, a hemolysis assay was performed using a 2% (v/v) erythrocyte suspension. First, 2 mL of freshly anticoagulated whole human blood was diluted with 5 mL of normal saline and centrifuged at $100 \times g$ for 15 min. The resulting erythrocytes were washed three times with normal saline, and then the pure erythrocytes were diluted to prepare a final 2% (v/v) erythrocyte suspension. Each sample was preheated in 0.8 mL of normal saline at 37 °C for 30 min. Next, 0.2 mL of the 2% (v/v) erythrocyte suspension was carefully added to the samples, followed by gentle mixing. The mixtures were incubated at 37 °C for 1 h, then centrifuged at $100 \times g$ to separate intact erythrocytes. The supernatant was cautiously transferred to new tubes for photographic documentation, and its absorbance was measured at 540 nm using a microplate reader (Multiscan™ FC Microplate Photometer, ThermoFisher Scientific, Waltham, MA, USA). Normal saline and deionized (DI) water served as the negative and positive controls, respectively. The hemolysis ratio was quantified using Eq. (15)[52].

$$\text{Hemolysis Ratio}(\%) = (V_H - V_{NC})/(V_{PC} - V_{NC}) \times 100 \tag{15}$$

In Eq. (15), $V_H$, $V_{NC}$, and $V_{PC}$ symbolize the absorbance of the supernatant corresponding to the samples (NA, MA, NMA, or XStat®), negative control, and positive control groups, respectively.

The hemostatic efficacy of NA, MA, and NMA was assessed by determining BCI, with QuikClot® Combat Gauze (QCG®) and XStat® used as comparative controls. For this evaluation, QCG® combat gauze and compressed shape-memory samples (e.g., NA, MA, NMA, and XStat®) were placed in EP tubes. Following a 10-min incubation at 37 °C, 50 μL of citrated whole human blood (UNMC Blood Bank) was carefully applied to the top surface of each sample. The samples were then incubated for an additional 10 min at 37 °C. Following this, 1.5 mL of DI water was added to each EP tube. The optical density of the resulting supernatant was measured at 540 nm ($OD_{540}$ nm) using a microplate reader (Multiscan™ FC Microplate Photometer, ThermoFisher Scientific), and this measurement was recorded as the Value of Hemostats ($VH_{BCI}$). A control solution of mixed DIW and CWB (1.5 mL/50 μL) was used to establish a baseline reference. The $OD_{540}$ nm value from this negative control solution served as the Value of Control

($VC_{BCI}$), while $VR_{BCI}$ defines the reference values for the subsequent calculations. The BCI was then computed using Eq. (16).

$$\text{Blood Clotting Index}(\%) = \{(VH_{BCI} - VR_{BCI})/(VC_{BCI} - VR_{BCI})\} \times 100 \tag{16}$$

We next determined the whole blood clotting time for NA, MA, and NMA in comparison to QCG® and XStat®. In this experiment, 500 µL of whole blood containing 10% sodium citrate was added to a polypropylene tube containing each sample (NA, MA, NMA, XStat®, and QCG®). Then, 20 µL of a 0.25 mol $L^{-1}$ $CaCl_2$ solution was added to the tube. The tube was inverted every 5 s, and the onset of blood clotting was recorded at each interval.

The interactions between the samples and RBCs were also investigated, with QCG® and XStat® serving as commercial controls. Compressed samples of NA, MA, NMA, and XStat® and QCG® were placed in a 24-well microplate. Next, 100 µL of the RBCs suspension was applied to the top surface of each sample. After a 1-h incubation at 37 °C, the samples were rinsed with a phosphate buffer solution (PBS, pH = 7.4) to remove non-adherent RBCs. The samples were then transferred into 4 mL of DI water to lyse the adhered RBCs and release hemoglobin. After another 1-h incubation, 100 µL of the supernatant was extracted and placed into a 96-well microplate, where its optical density at 540 nm ($VH_{RBC}$) was measured. The $OD_{540}$ nm value of a solution containing 100 µL of RBCs suspension and 4 mL of DIW was used as the control value ($VC_{RBC}$), while $VR_{RBC}$ defines as reference values. The percentage of adhered RBCs was calculated using Eq. (17).

$$\text{RBC Adhesion}(\%) = \{(VH_{RBC} - VR_{RBC})/(VC_{RBC} - VR_{RBC})\} \times 100 \tag{17}$$

A platelet adhesion assay was conducted to further investigate the interactions between various hemostats and platelets. To prepare for this assay, platelet-rich plasma (PRP) was obtained by centrifuging citrated whole blood at $100 \times g$ for 15 min. Similar to the RBC assay, NA, MA, NMA, and XStat® were compressed, placed in a 24-well microplate, and 100 µL of PRP was applied to their top surfaces. For QCG®, 100 µL of PRP was applied on the top of it. After incubating the samples for 1-h at 37 °C, they were washed with PBS to remove non-adherent platelets. The samples were then soaked in a 1% Triton X-100 solution to lyse the adhered platelets and release the lactate dehydrogenase (LDH) enzyme. The optical density at 490 nm ($OD_{490}$ nm) of the resulting supernatant was measured ($VH_{Plt}$). The $OD_{490}$ nm value of a solution containing 100 µL of PRP that had not been exposed to the hemostats was used as the control value ($VC_{Plt}$), while $VR_{Plt}$ was defined as reference values. The percentage of adhered platelets was calculated using Eq. (18).

$$\text{Platelets Adhesion}(\%) = \{(VH_{Plt} - VR_{Plt})/(VC_{Plt} - VR_{Plt})\} \times 100 \tag{18}$$

To further investigate the adherence of RBCs and platelets on various hemostats, a detailed SEM (FEI Quanta 200, Hillsboro, OR, USA) analysis was conducted. In this analysis, samples of NA, MA, NMA, and XStat® were placed in individual wells of a 24-well microplate and exposed to 100 µL of RBCs and PRP suspensions. After a 1-h incubation at 37 °C, the samples were rinsed with 1×DPBS and fixed with a solution containing 2% paraformaldehyde and 2.5% glutaraldehyde (Ladd Research). The samples were thorough washed, treated with osmium tetroxide (Sigma-Aldrich), and then washed again. They were dehydrated using a graded ethanol series, and subsequently treated with a graded series of HMDS (Sigma-Aldrich). Finally, the samples were dried in 100% HMDS within a chemical hood.

To investigate the coagulation activation pathway of NA, MA, and NMA, as well as the commercially available products XStat® and QCG® (which served as positive controls), PT and aPTT assays were conducted. Citrated whole human blood was prepared by mixing whole blood with 3.8% sodium citrate in a 9:1 ratio, followed by centrifugation at $1500 \times g$ for 15 min to obtain platelet-poor plasma (PPP). For the PT assay, each sample was incubated with 200 µL of PPP at 37 °C for 5 min. After incubation, 200 µL of PT reagent (Fisher Scientific) was added to the hemostats, and the time required for PPP to clot was recorded using a stopwatch.

Similarly, the aPTT assay involved incubating of each sample with 200 µL of aPTT reagent (Fisher Scientific) at 37 °C for 5 min. Following this, 100 µL of a 0.025 M $CaCl_2$ solution was added to activate the intrinsic coagulation pathway, and the clotting time taken was recorded using a stopwatch (Samsung Galaxy Note 20 Ultra).

## Weighted ranking of aerogels

A weighted ranking methodology was used to select the best-performing aerogel samples for in vivo experiments[53]. First, a Z-score transformation was applied to the raw data to standardize and normalize the datasets, making the results easier to interpret. The Z-score scale, with a mean of '0' and a standard deviation of '1', allowed for robust comparisons among NA, MA, and NMA. Each aerogel's performance metrics were quantified and standardized using the Z-score formula, which involves subtracting the sample mean (µ) from the observed value (x) and then dividing by the sample standard deviation (σ), as delineated in Eq. (19).

$$Z = \frac{x - \mu}{\sigma} \tag{19}$$

Moreover, categorical and subjective weighting was applied to two critical criteria−physical characteristics and hemostatic properties −to refine the analysis. Physical characteristics included factors such as porosity, percentage of open pores, specific elastic modulus, shape memory, blood absorption rate, and the adhesion of RBCs and platelets. Hemostatic properties included variables like shape recovery time in human blood, BCI, blood clotting time, PT, aPTT, and hemolysis.

To provide a more precise evaluation of each aerogel's overall performance, weightings were assigned to each parameter based on its relative importance. These weightings were determined through domain expertise and a detailed understanding of their significance. By combining these weighted factors with standardized Z-scores, each aerogel was ranked according to its performance, making it easier to identify the optimal designs. This approach facilitated direct comparisons among the various tested aerogels with different units, offering a clear framework for optimizing aerogel design for subsequent investigations. The weighted average was computed using Eq. (20),

$$\bar{x} = \frac{\sum_{i=1}^{n} w_i \cdot x_i}{\sum_{i=1}^{n} w_i} \tag{20}$$

Where x represents the weighted average, $w_i$ signifies the sum of the product of the weight, and $x_i$ denotes the data number.

## Animal studies

The experimental design for the animal study adhered to the scientific rigor and transparency required by the National Institute of Health (NIH) and followed the ARRIVE guidelines (Supplementary Table 8)[32]. Approval was obtained from UNMC IACUC under protocol no. 22-051-08-EP, with all procedures conducted in strict compliance with UNMC IACUC guidelines.

The study involved inbred Yorkshire swine (3 months, both male and female, all non-castrated, $n = 5$), sourced from a UNMC-approved vendor. The animals were fasted for 12-h before surgery, with free access to water. On the day of surgery, premedication was administered, and an intravenous (IV) line was established in a marginal ear vein using a 20−22-gauge Angiocatheter (Nordson Medical, Loveland,

CO 80538, USA). Anesthesia was induced via isoflurane and oxygen (3–5 L/min) using a veterinary anesthesia ventilator (MDS Matrx 3000, HALLOWELL EMC, Pittsfield, MA 01201, USA) to facilitate intubation. During the procedure, anesthesia was maintained via isoflurane and oxygen (1–2 L/min) until death or euthanasia. Vital signs were continuously monitored, with a rectal temperature probe and EKG (cardiac) monitors (DRE Waveline VS, M16C13140003, PET PRO SUPPLY CO.®, Frisco, TX, USA) in place. The swine were positioned on a water-circulated warming blanket (BLANKETROL® II, 222R, Cincinnati Sub-Zero Products, Inc., Cincinnati, OH 45241, USA), maintained at 103 °F. Mechanical ventilation was set at 12–15 breaths per min with a tidal volume of 5–10 mL/kg (MDS Matrx 3000, HALLOWELL EMC, Pittsfield, MA 01201, USA), and end-tidal $pCO_2$ was maintained between 35–45 mmHg. All incisions were made using an electrosurgical generator (Valleylab Force 1C, Pfizer, New Bedford, MA, USA).

For pressure monitoring and blood sampling, a 20-gauge Angiocatheter was surgically placed in the carotid artery and a 14–16-gauge Angiocatheter was inserted in the jugular vein for fluid and medication administration. These catheters (Nordson Medical, Loveland, CO 80538, USA) were inserted through a surgical cutdown in the right or left neck (Supplementary Fig. 18). Concomitantly, a midline laparotomy and splenectomy were performed to minimize autotransfusion from the contractile porcine spleen during stress[49], a mechanism that is not present in humans but crucial for a human-like hemorrhage model (Supplementary Figs. 19, 20 and Supplementary Movie 12). The spleen was weighted (Supplementary Fig. 21), and warm lactated Ringers (LR) solution was administered at three times the spleen's weight at a rate of 100 mL/min using a roller infusion pump (DOSE IT P910, Integra Biosciences AG, Tardisstrasse, Zizers 7205, Switzerland). For a typical spleen weighing ~300 g, approximately ~900 mL of LR was given for post-splenectomy fluid replacement. A transabdominal cystostomy tube was placed through the bladder dome, secured with a purse-string suture, and exited through the lateral abdominal wall (Supplementary Fig. 22 and Supplementary Movie 13). This procedure controlled urine flow during the experiment and prevented undue pressure on the injury site, which could affect the hemorrhage rate. Afterward, the midline laparotomy incision was closed using towel clips, and the animals were covered with blankets to prevent hypothermia due to post-injury blood loss (Supplementary Movie 14).

To perform the surgical incision and identify the right femoral artery in domestic pigs, a midline incision was made in the right groin area using an electronic scalpel, exposing the underlying tissues. The right femoral artery and vein were chosen due to their diameter, which closely resembles that of human vessels, ensuring that the bleeding force in the swine closely mimicked human conditions. Subsequently, the subcutaneous tissue in the pig was carefully dissected until the femoral vessels were visually identified (Supplementary Fig. 23 and Supplementary Movie 15). The right femoral artery and vein were then clearly identified, and a deliberate injury to these blood vessels was induced by surgically transecting the femoral artery and vein under general anesthesia. This procedure allowed for a controlled 30-s period of free bleeding, simulating the conditions of a junctional injury similar to what might be encountered in a warfighter, such as a gunshot wound to the groin. The experimental design included four groups, each consisting of five pigs (the sample size was chosen according to a power analysis), to systematically investigate the impact of different treatments on the induced injury:

1. Injury Only (Non-Treated Control): This group serves as the non-treated control, where no specific treatment is applied to the induced injury. The purpose is to showcase the inherent lethality of an uncontrolled injury, providing a baseline for comparison with treated groups.
2. Injury with XStat® Treatment: Inclusion of this group involves administering the current primary comparator treatment for JH,

utilizing XStat® pellets, in accordance with Tactical Combat Casualty Care (TCCC) guidelines[7]. This treatment is established as the standard for comparison due to its recognized effectiveness.
3. Injury with QCG® Treatment: This group functions as a secondary comparator treatment, employing QCG®–a widely available standard issue in the U.S. Army. It represents an alternative intervention for JH, especially when XStat® is unavailable, providing a practical comparison to the primary treatment.
4. Injury with NMA Treatment: This group constitutes the primary experimental treatment, involving the application of dried bicomponent aerogel pellets. The purpose is to explore the efficacy of the shape memorable NMA as an experimental intervention for JH. This treatment offers a unique approach, distinct from wet or biologically augmented treatments in the published reports, allowing for a comprehensive evaluation of its potential effectiveness.

Two devices of each type such as QCG®, XStat®, or NMA were applied to the injury site. For NMA, each device contains 35–50 compressed pellets, whereas each device of XStat® contains ~100 pellets. The application times of each application material were recorded using a stopwatch and the effects of these treatment groups were meticulously evaluated in terms of primary and secondary endpoints during a 180-min observation period conducted under general anesthesia. The selection of the 180-min time point was based on the time required for transferring patients to hospitals. Primary endpoints included the assessment of post-treatment blood loss, survival status at 180 min post-injury, and the incidence of post-treatment rebleeding.

The blood collected during or after compression, including any bleeding during the compression period, was directed into a separate suction canister. This separation facilitated the differentiation of blood loss before and after treatment. Following treatment, a new set of surgical gauze was employed for compression over a 3-min, during which the gauze absorbed blood. The manual pressure was applied with adequate force to stop the bleeding. Specifically, for this groin injury model, pressure was applied using both hands: the right hand applied direct and deep pressure on the hemostatic material, while the left hand pushed on top of the right hand to reinforce the force. Although there is no standardized or published methodology to precisely calibrate hand pressure in a model such as this, the combined force was likely in the range of 15–20 lb based on the estimation of our surgeons. Additionally, the application materials played a role in blood clotting by absorbing the whole blood within the wound. Considering the collective blood absorption by both the application materials and the surgical gauzes during or after compression as part of the post-treatment blood loss, the following Eq. (21) was employed for accurate quantification.

$$\text{Post Treatment Blood Loss} = (WC_{b'} - WC_e) + (WG_{w'} - WG_d) + (WM_b - WM_d)$$
(21)

Where $WC_{b'}$ represents the weight of the canister used to collect blood lost after treatment until death or after 180 min of observation, and $WC_e$ denotes the empty weight of the canister. $WG_{w'}$ is the weight of wet gauze used for post-treatment manual compression for 3 min, and $WG_d$ is the dry weight of the gauze. $WM_b$ represents the weight of application materials after treatment, and $WM_d$ is the weight of the materials before treatment.

The hemostasis percentages for each group depict the proportion of animals that ceased bleeding within a specified time frame, as determined by Eq. (22).

$$\text{Hemostasis}(\%) = (N_H/n) \times 100$$
(22)

Here, Eq. (22) defines $N_H$ as the number of swine stopped bleeding after treatment, while "n" represents the total number of experimental

animals in the group. The hemostatic time for each application material was measured using a stopwatch, representing the total time each material took to stop bleeding after treatment. Rebleeding time, defined as the cumulative time of all rebleeding incidences for each subject after treatment, was recorded using a stopwatch. The percentage incidence of rebleeding in each group was calculated using Eq. (23).

$$\text{Rebleeding Incidence}(\%) = (N_{rb}/n) \times 100 \qquad (23)$$

In Eq. (23), $N_{rb}$ represents the total number of experimental animals experiencing post-treatment rebleeding, while 'n' represents the total number of experimental animals in the group.

The hemostatic time and rebleeding percentages following treatment with NMA were systematically compared with relevant literature from 2010–2024. Our search encompassed databases such as Google Scholar, PubMed, and Scopus, focusing on studies on swine or pig artery and/or vein transection hemorrhage models. The search terms included swine, pig, hemostatic agent or materials, and hemostasis in swine artery and/or vein transection models. The findings from these searches were compiled and summarized in Supplementary Table 9, detailing the referenced papers' experimental designs, models, and conditions. This table elucidates the comparability of these reference papers with the present study. Specifically, it highlights instances where the models and experimental conditions of the reference papers align with or are even less aggressive than those employed in this study.

Secondary endpoints included the assessment of final vital signs (i.e., mean arterial pressure and heart rate), hematologic parameters (e.g., hemoglobin, hematocrit, platelet count, and lactate levels), coagulation profiles (e.g., prothrombin time, partial thromboplastin time, international normalized ratio [INR], and fibrinogen levels), and various arterial blood gas parameters (e.g., bicarbonate [$HCO_3$], partial pressure of oxygen [$PaO_2$], partial pressure of carbon dioxide [$PaCO_2$], and end-tidal carbon dioxide [$EtCO_2$]), as well as pH and temperatures.

Mean arterial pressure, heart rate, and body temperatures were monitored by a designated technician from UNMC Comparative Medicine, who was blinded to both samples and subjects at predetermined time intervals during the experiments (e.g., 0 min, 15 min, 30 min, 60 min, 120 min, and 180 min) using an in-house EKG monitor (DRE Waveline VS, M16C13140003, PET PRO SUPPLY CO., Frisco, TX, USA).

Hematologic, coagulation, and arterial blood gas analyses were conducted at UNMC Hospital Pathology by pathologists unaware of the study details. Complete blood counts were performed using a Sysmex XN 9100 instrument (Sysmex America, Mundelein, IL, USA) at specified time points during the experiment (0 min, 15 min, 30 min, 60 min, 120 min, and 180 min). Lactate levels were measured using a Beckman Coulter AU5800 (Beckman Coulter Inc., Higashino, Shizuoka, Japan) machine at the same time intervals. Arterial blood gas values were assessed using a Radiometer ABL800 FLEX Analyzer (Radiometer Medical ApS, Aakandevej, Bronshoj, Denmark) at UNMC Hospital Pathology to measure $HCO_3$, $PaO_2$, $PaCO_2$, and $EtCO_2$ at various time points during the experiments. Coagulation parameters, including prothrombin time, partial thromboplastin time, INR, and fibrinogen levels, were determined using ACL TOP 500 instruments (Instrumentation Laboratory Company, Bedford, MA, USA) at different time intervals throughout the study. As mentioned above, the sample size was $n = 5$ for the Control, NMA, XStat®, and QCG® groups. However, for the baseline measurements of lactate level, the sample size was $n = 4$ for the Control, QCG®, and NMA groups due to insufficient sample quantity, instrumental error, or clotting of the blood prior to measurement (Supplementary Fig. 16). Similarly, the sample size was $n = 4$ for the Control group (for the baseline measurements of PT, aPTT, INR, and fibrinogen) and QCG® (for the baseline measurements

of PT, aPTT, and INR) due to the same limitations (Supplementary Fig. 17).

In instances where a subject succumbed to hemorrhage before the 180-min mark, final secondary endpoints were obtained during the pre-terminal phase. Following the designated observation period, each pig underwent euthanasia through the intentional transaction of a large blood vessel in the chest, facilitating terminal hemorrhage (Supplementary Movie 16).

## Data adoption for comparison

The data for the control, XStat®, and QCG® groups were adopted from our recent report to compare with the NMA treatment group[1]. This method was selected to adhere to the 3Rs principle of animal research by minimizing the number of animals used while maintaining robust control data, as the data were collected under the same experimental protocol[54,55].

## Power analysis

The sample size for the in vivo study was determined using power analysis, aimed at assessing survivability in a non-survival JH swine model. We hypothesized that without treatment, 90% of subjects would perish, whereas with NMA treatment, 90% would survive, with a 10% mortality rate. Based on an expected mortality rate of 10% in Group 1 (no treatment, control) and 90% in Group 2 (NMA treatment), with an alpha level of 0.05 and a beta level of 0.2 (power of 0.8), the necessary sample size was calculated. The sample size was determined using a two-sided test for comparing two proportions, specifically to detect a clinically significant difference in survival rates between the untreated and NMA-treated groups[56]. The calculation was based on the following equation:

$$n = \left\{ z_1 - \alpha/2 \times \sqrt{\bar{p} \times \bar{q}} \times \left(1 + \frac{1}{k}\right) + z_1 - \beta \times \sqrt{\bar{p} \times \bar{q} + \left(\frac{p_2 \times q_2}{k}\right)} \right\}^2 / \Delta^2$$

(24)

In this Eq. (24), the proportion (incidence) of subjects in groups #1 and #2, denoted as $p_1$ and $p_2$, respectively. Additionally, the absolute difference between these proportions, $\Delta$, was calculated to quantify the expected disparity in outcomes between the two groups. The probabilities of type I and type II errors, denoted as $\alpha$ and $\beta$, respectively, were set at standard levels of 0.05 and 0.2. Critical Z values corresponding to these error probabilities were utilized in the sample size calculation. Moreover, the ratio of sample size between group 2 and group 1, denoted as "$k$", was considered to ensure balance and efficiency in the study design. The values of $q_1$, $q_2$, $\bar{p}$, $\bar{q}$, and $\Delta$ were calculated using the respective equations.

$$q_1 = 1 - p_1 \qquad (25)$$

$$q_2 = 1 - p_2 \qquad (26)$$

$$\bar{p} = \frac{p_1 + kp_2}{1 + k} \qquad (27)$$

$$\bar{q} = 1 - \bar{p} \qquad (28)$$

$$\Delta = |p_2 - p_1| \qquad (29)$$

Based on these parameters, the analysis determined that a sample size of 5 animals per group was required to achieve an alpha level of 0.05 and a beta level of 0.2 (power of 0.8). The in vivo study included four groups: Control (no treatment), XStat®, QCG®, and NMA.

## Statistical analysis

The Shapiro–Wilk test was used to assess data normality, and the data were presented as mean values with corresponding standard deviations (s.d.). Statistical analyses were conducted utilizing analysis of variance (ANOVA) with GraphPad Prism (version 9.5.1). Pairwise comparisons were carried out using either ordinary one-way or two-way ANOVAs, followed by Tukey's multiple comparisons post hoc test when appropriate. Additional statistical comparisons were performed as indicated. For measurements obtained from photographs or SEM images, Image J was utilized after calibrating pixels to millimeters or micrometers. Survival analysis was performed using the log-rank tests to compare survival curves among groups. The Fisher exact test was employed for categorical data comparisons to assess statistical significance. All analyses were conducted with a significance level set at 0.05. Graphs and figures were created and visualized using GraphPad Prism, Origin Pro (version 8.5), and BioRender.

## Reporting summary

Further information on research design is available in the Nature Portfolio Reporting Summary linked to this article.

## Data availability

Source data are provided with this paper.

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

## Acknowledgements

This work was partially supported by startup funds from the University of Nebraska Medical Center (UNMC) (J.X.), National Institute of General Medical Science (NIGMS) of the National Institutes of Health under Award Number R01GM138552 (J.X.), Congressionally Directed Medical Research Program (CDMRP)/Peer Reviewed Medical Research Program (PRMRP) W81XWH2010207 (J.X.) and W81XWH2010208 (M.A.C.), Nebraska Research Initiative Grant (J.X.), and NE LB606 (J.X.).

## Author contributions

S.M.S.S., S.M.A., and A.A. contributed equally. S.M.S.S., S.M.A., A.A., Z.Y., A.D.M., C.W., Z.A.Y., N.S.S., M.E.B., M.I.N., and C.P.J. performed the studies at UNMC. S.M.S.S., S.M.A., and A.D.M. characterized materials. S.M.S.S., S.M.A., and A.D.M. conducted, analyzed, and interpreted mechanical properties. Z.Y. performed simulation studies. A.A. and M.A.C. performed in vivo studies. S.M.S.S., S.M.A., and C.P.J. assisted in surgery. S.M.S.S., S.M.A., C.W., Z.A.Y., M.E.B., and M.I.N. fabricated the materials. K.Y., M.A.C., and J.X. provided critical intellectual input and contributed to the study design and interpretation of results. S.M.S.S., S.M.A., A.A., A.D.M., M.A.C., and J.X. wrote and revised the manuscript with input from all authors.

## Competing interests

The authors declare no competing interests.
