## [Peer review file · Nature Communications]

Bicomponent nano- and microfiber aerogels for effective management of junctional hemorrhage

Corresponding Author: Professor Jingwei Xie

Version 0:

Reviewer comments:

Reviewer #1

(Remarks to the Author)

Non-compressible bleeding at the junction has long been an important factor in battlefield and civil trauma deaths. In this study, a kind of aerogel material consisting of two component (nanofibers and microfibers) is reported. The material is composed of polylactic acid nanofibers, polycaprolactone microfibers and gelatin. The aerogel formed by nanofibers, microfibers and glutaraldehyde crosslinked gelatin has good mechanical properties and shape recovery. When injected into a junctional wound, the aerogel can quickly stop bleeding by expanding and squeezing, and promoting clotting in a synergistic manner. The author fully studied its mechanical properties, morphology, shape memory, procoagulant activity and hemostatic effect in vivo, and used commercial Xstat as a control. The hemostatic performance of this material is quite outstanding. The design of this material is relatively novel, and it has great potential for non-compressible bleeding at the junction, which is of great significance in this field. However, the following important issues still need to be resolved.

1. The introduction part pointed out that "involving complex and costly manufacturing processes, which challenge large-scale production". However, in my opinion, the preparation method of these aerogels presented in the article does not have the advantages of low price and low cost.
2. In this article, a series of characterization and summary of the mechanical properties of the three aerogels are carried out. However, the explanation of the implementation principle and mechanism behind them is still less, please supplement this part.
3. The survival rate of pigs after NMA haemostasis was 100%, what treatments were carried out after successful NMA haemostasis?
4. What are the criteria for setting the number of different groups of materials used in the article? At present, the number of materials used in each group varies greatly. This has a great effect on the hemostatic performance of the material.
5. What is the uniformity of the micropores after the preparation of the material? In addition, when there is a large number of preparation needs, can you prepare a large area of material and then cut? At present, the material in this article is prepared one by one, which is not conducive to production.
6. The author performed hand compression during hemostasis, but the strength of the compression was not given in the article. The addition of this data is conducive to the comparative evaluation of hemostasis effect for readers.
7. The hemostatic efficiency of the product Xstat used in the study was not good, and no complete hemostasis was achieved in the five repeats. However, the hemostatic efficiency of this material in other studies is much higher than 80 percent, how do the authors explain this phenomenon?
8. How persistent is the shape memory performance of the material during long-term storage? With the extension of storage time, will its shape memory effect decrease significantly?
9. One of the advantages of this material is that it can be degraded. However, the whole article did not do any test on degradation, which needs to be added.
10. The author uses PLA to prepare electrospun fibers, but uses PCL to prepare microfibers. What is the basis for choosing these two materials? In addition, the screening process for the diameter of the fiber material and the length of the short fiber is not given. These parameters have a significant effect on the mechanical properties and hemostatic properties of the material. How does the author take this into account?
11. Although the material can be degraded, it is widely used in hemostasis and still needs to be removed later. Therefore, the tensile strength of the material can significantly affect its removal performance. The authors need to give the tensile strength test of the material and compare it with the Xstat material. In addition, under the premise of heavy use, how can you

ensure that every aerogel in the wound is completely removed, thereby reducing its adverse effect on wound healing?

12. What is the shape memory and shape recovery mechanism of the material?

13. Biosafety evaluation is an important prerequisite for material application. The biocompatibility of the material, such as cell viability and inflammatory response during implantation in vivo, was not tested in this article, which need to be supplemented.

14. Some important parameters in the process of materials' preparation are still relatively lacking, such as the molecular weight of PLA and PCL, and the cutting length of short fibers. Complete and clear material preparation parameters are conducive to readers' reference.

Reviewer #2

(Remarks to the Author)

Reviewer #3

(Remarks to the Author)

In this manuscript, the author developed bicomponent aerogels based on poly (lactic acid) (PLA) nanofibers and poly(ϵ -caprolactone) (PCL) microfibers for effective halting bleeding on junctional hemorrhage. The aerogel exhibited excellent absorption capacity and in vitro blood clotting efficacy. This work is interesting, well-structured, and logical. Therefore, I recommend it has the potential for publication in Nature Communications after major revisions.

Some specific comments:

1. The whole manuscript has no line and page numbers.

2. The first three sentences of the Introduction section cite a total of 20 references, and I think this is an overquoting phenomenon.

3. The abbreviations of NMA need to be defined in the first appearance.

4. "Due to its high absorption properties, the PLA NF component could effectively entrap platelets and clotting factors and enhance the aggregation of red blood cells (RBCs)." "Due to its high absorption properties" is repeated twice.

5. Are there long-term follow-up studies to evaluate the hemostatic efficacy and safety of aerogels, including their biodegradation process, long-term safety, and potential impacts on surrounding tissues?

6. Although the article mentions that the material could potentially be used in battlefield or emergency rescue situations, not provide any experimental data or discussion regarding the material's performance under harsh environmental conditions, such as extreme temperatures, humidity, or pressure.

7. The study conducted comparative experiments between the aerogel and XStat®, demonstrating the aerogel's advantages in shape memory and absorption capacity. However, the specific differences in the hemostatic mechanisms between the aerogel and traditional hemostatic materials were not mentioned.

8. When conducting the aerogel compression test, the author should show the actual image of the aerogel in the test process in Figure 2, so as to show the ability of shape recovery of the aerogel more intuitively.

9. The author should briefly describe the chemical properties of PLA and PCL for understanding the role of PLA and PCL in gelatin-based aerogels.

10. Do aerogels have antibacterial properties? Whether the author consider adding antibacterial agent to the aerogels to prevent infection at the junctional wound site?

11. Will the aerogel remain partially in the wound after use? Authors should determine the water contact angle to investigate wettability of aerogel, and observe the integrity of the aerogels after fully absorbing blood.

12. "The compressed NMA pellets could be injected into the deep and narrow wounds, where they might re-expand to form a tamponade and stop the bleeding". The manuscript stated that NMA pellets are injected into wounds, but the Supplementary Materials stated that NMA pellets are submerged in wounds. Is it injection or submersion? The compressed NMA pellets Size?

13. "Additionally, pore diameter analysis revealed values of $236.70 \pm 58.03 \mu\text{m}$ and $183.10 \pm 48.98 \mu\text{m}$ for MA and HA", what is HA?

14. The format of the references needs to be checked carefully.

15. The manuscript requires English revision. There are detailed mistakes throughout the manuscript that need to be corrected.

Reviewer #4

(Remarks to the Author)

What are the noteworthy results?

NMA is superior to QCG and Xstat (established hemostatics) in junctional hemorrhage.

Will the work be of significance to the field and related fields? How does it compare to the established literature? If the work is not original, please provide relevant references.

The authors describe a novel biogel, that has hemostatic properties, and improved survival and hemostasis in a porcine model of uncontrolled femoral hemorrhage. Overall, the study is interesting, and novel materials to produce hemostasis from uncontrolled junctional hemorrhage are needed. The work may be of benefit to civilian and military patients with junctional haemorrhage.

Does the work support the conclusions and claims, or is additional evidence needed?

There are some shortcomings that need to be discussed, below.

Are there any flaws in the data analysis, interpretation and conclusions? Do these prohibit publication or require revision?

There is no clearly stated hypothesis and research question to be answered. Even though there seems to be answers to a question. Please state this more clearly in the introduction.

The statistical analyses are sound. One- and two-way ANOVA with Tukey's multiple comparisons test seem correct analyses for the data (depending on distribution). Survival analysis using log-rank and Fisher also seem correct for the categorical data.

Information on data distribution is missing. Did you perform Smirnov-Kolmogorov or Shapiro-Wilks to test for normality?

Statistical analyses are only provided in the figure legends. It would be good to see a dedicated "statistical analysis" section in the method part. The graphical work is clear and readable.

Is the methodology sound? Does the work meet the expected standards in your field?

How did you decide on the number of animals in each group, n=5? Did you perform an a priori power calculation? How was the allocation of treatment elected? Randomly? Non-randomly? Was any part of the study blinded? All should be included according to the ARRIVE guidelines. What was the average weight of the pigs? How were the pigs sedated, medications and doses? How were they monitored? Where did you measure arterial blood pressure? A methodology section is missing. The rationale and explanation of the haemostatic agents are extensive and informative. However, the methodology section needs improvement, although the movie material is informative.

Is there enough detail provided in the methods for the work to be reproduced?

It would probably be possible with all the informative supplemental material. The methodology section needs improvement. Suppl fig 21. Was randomization between treatments performed?

The authors performed a splenectomy before hemorrhage, and a laparotomy. This is invasive and may confound results in terms of hemostasis, inflammation and more. Please comment on the rationale and limitations. Splenectomy before bleeding is not always necessary (PMID: 26974424).

How did you measure post-treatment blood loss accurately? Often, the exact amount of blood coming from a junctional area of free bleeding, especially when some material was applied, may be challenging. It is likely that some of the blood in the area was coagulated. Or was it not?

The level of accuracy is artificially high. For example, the authors report that "the average blood loss, including the blood absorbed by the hemostasis, was only 42.908 mL". How did you measure the blood loss from a free bleeding model with this accuracy? It is inherently difficult to measure the exact bleeding in free bleeding models. Moreover, even if the .908 mL were truthfully accurate, please discuss the clinical relevance of this accuracy.

"Additionally, all untreated animals expired". This is an unnecessary euphemism. Expired means died.

Coagulation tests are missing, and data on how standard coagulation was affected by the treatment.

Reviewer #5

(Remarks to the Author)

Version 1:

Reviewer comments:

Reviewer #1

(Remarks to the Author)

I appreciate the thoroughness and clarity with which you addressed the issues raised in my review. Your revisions have resolved my concerns. Additionally, I have reviewed your responses to the reviewer 2's comments and agree that the concerns have been adequately addressed. The manuscript can be accepted in recent form.

Reviewer #3

(Remarks to the Author)

After revision, the authors addressed all the questions I proposed well. Therefore, I do recommend that this manuscript can be accepted for publication in Nature communications

Reviewer #4

(Remarks to the Author)

The authors have successfully answered the questions, and have revised the manuscript accordingly. I have no further questions.

Reviewer #5

(Remarks to the Author)

Response to Referees

Reviewer #1 (Remarks to the Author):

Introductory Statement: *Non-compressible bleeding at the junction has long been an important factor in battlefield and civil trauma deaths. In this study, a kind of aerogel material consisting of two component (nanofibers and microfibers) is reported. The material is composed of polylactic acid nanofibers, polycaprolactone microfibers and gelatin. The aerogel formed by nanofibers, microfibers and glutaraldehyde crosslinked gelatin has good mechanical properties and shape recovery. When injected into a junctional wound, the aerogel can quickly stop bleeding by expanding and squeezing, and promoting clotting in a synergistic manner. The author fully studied its mechanical properties, morphology, shape memory, procoagulant activity and hemostatic effect in vivo, and used commercial Xstat as a control. The hemostatic performance of this material is quite outstanding. The design of this material is relatively novel, and it has great potential for non-compressible bleeding at the junction, which is of great significance in this field. However, the following important issues still need to be resolved.*

Author's Response: We sincerely thank Reviewer #1 for the positive and encouraging feedback regarding our study. We greatly appreciate your recognition of the novelty and potential impact of our materials for addressing non-compressible bleeding at the junction. The reviewer's comments highlight the importance of our work, and we are grateful for his/her thoughtful and constructive suggestions. We have carefully addressed all the concerns and questions the reviewer raised in the following.

Specific Comments:

1. *The introduction part pointed out that "involving complex and costly manufacturing processes, which challenge large-scale production". However, in my opinion, the preparation method of these aerogels presented in the article does not have the advantages of low price and low cost.*

Response: We agree with the reviewer's insightful comment. We have made changes to this sentence in the Introduction section. (See revised manuscript, Page 3).

The revision is as follows: "However, many potential treatments currently under development have one or more following characteristics that could affect their utilities, such as: (i) remaining in a wet state, which is unsuitable for blood concentration, and long-term storage, and may lead to fungal contamination; (ii) requiring the incorporation of clotting factors, necessitating special storage conditions that are impractical for battlefield settings, and risk degradation during transportation; (iii) exhibiting poor mechanical properties and slow expansion; (iv) lacking a safe and effective field delivery system; and (v) being non-resorbable, necessitating complete removal."

2. *In this article, a series of characterization and summary of the mechanical properties of the three aerogels are carried out. However, the explanation of the implementation principle and mechanism behind them is still less, please supplement this part.*

Response: Thanks for the valuable comment. In addition to the **Results section (Pages 6-7)**, where we clearly highlighted the importance of each mechanical test, we have added a dedicated description to the Discussion section. (See revised manuscript, Pages 14-15) This section further explains the significance of the extensive mechanical property analysis, supported by our results and existing literature.

The discussion section is as follows: "Materials used in junctional wounds for hemostasis must have strong elastic properties to re-expand and fill the cavities, creating a tamponade effect. The only FDA-approved shape expandable hemostat, XStat®, and many other products under development for managing JH might exhibit suboptimal mechanical properties and structures, which could potentially limit their effectiveness in hemostasis. The bicomponent aerogel mainly consisted of PLA nanofibers and PCL microfibers, which entangle, forming a highly porous structure with a high percentage of open pores. The aerogel demonstrated excellent mechanical properties, including a high specific elastic modulus, high resilience, rapid shape recovery, and exceptional robustness. These attributes were likely due to its intricate network composed of dual-sized fibers, along with gelatin coating and crosslinking. Optimizing the elastic energy release from shape-expandable hemostatic pellets is crucial. Insufficient compression modulus could result in a slow expansion rate, worsening blood loss from the wound, while excessive compression modulus could exert severe pressure, leading to additional soft tissue injury during application. In this study, the elastic modulus measurements under different compression rates demonstrated that NMA could recover

its shape effectively even after 90% compression. Interestingly, the compression modulus of NMA (ranging from 7.87 ± 2.81 kPa to 961.85 ± 48.69 kPa) was found to be lower than those of human soft tissues (ranging from several kPa to several MPa). Notably, the mechanical strength of the aerogels can be finely tuned by adjusting crosslinking time without compromising their resilience. This highlights the versatility of these materials in achieving the desired balance between compression strength and elasticity. Additionally, the 100-cycle compression-relaxation test demonstrated excellent fatigue resistance and mechanical stability. These properties are crucial for maintaining a strong and stable hemostatic barrier, which applies consistent pressure on surrounding tissues to effectively control bleeding.”

3. The survival rate of pigs after NMA haemostasis was 100%, what treatments were carried out after successful NMA haemostasis?

Response: NMA samples were injected, and manual compression was applied for 3 min to achieve hemostasis. Following this, the subjects were observed for the next 3 h without any further interventions to assess survival.

4. What are the criteria for setting the number of different groups of materials used in the article? At present, the number of materials used in each group varies greatly. This has a great effect on the hemostatic performance of the material.

Response: We understand the concern regarding the varying groups of materials used in different experiments. The variation is based on the specific objectives of each test in this study. For the aerogel's performance evaluation, we selected three groups—nanofiber aerogels (NA), microfiber aerogels (MA), and a combination of nano- and microfiber aerogel (NMA)—to compare the effectiveness of single-fiber versus dual-scale aerogels in halting bleeding. This comparison helps assess which material type performs better. For testing mechanical properties and shape memory, we chose NA, MA, and NMA aerogels along with XStat® as controls. QCG® was excluded from this experiment because it does not possess shape-memory properties, which are essential for this specific analysis. In the procoagulant activity test, QCG® was included because of its known blood coagulation properties, allowing us to assess the procoagulant activity of our aerogels in comparison to a clinically relevant product. For the in vivo studies, NMA was selected among three different aerogel groups based on its promising results in physical characterization, mechanical properties, shape memory, and procoagulant activity, as outlined in our weighted matrix. **(See revised manuscript, Pages 9-11 and Supplementary Tables 11-14)** XStat® and QCG® were included as current clinical treatments for comparison.

Regarding the variation in the number of materials used in the in vivo experiments, this was essential for targeted evaluation. NMA aerogels had 35-50 pellets per device, whereas XStat® had nearly 100 pellets per device. **(See revised manuscript, Page 33)** This difference highlights NMA's more efficient performance in certain tests, such as requiring fewer pellets compared to XStat®. This approach allowed us to conduct a more comprehensive analysis of the material properties while also considering practical aspects, such as cost-effectiveness and field applicability.

5. What is the uniformity of the micropores after the preparation of the material? In addition, when there is a large number of preparation needs, can you prepare a large area of material and then cut? At present, the material in this article is prepared one by one, which is not conducive to production.

Response: We appreciate the reviewer for raising this important point. The uniformity of the micropores in the material was analyzed using both SEM and micro-CT. **(See revised manuscript, Page 5)** These methods provided precise and accurate data on pore structure. Both analyses consistently demonstrated that the micropores are uniform, with the revised manuscript presenting the average micropore size and distribution from each batch. **(See revised manuscript, Pages 44-45 and S108, Fig. 1e, 1f, 1i, and 1j and Supplementary Fig. 3).**

Additionally, we would like to clarify that the materials in this study were not prepared one by one. Instead, we utilized molds designed to produce multiple samples simultaneously. We have discussed this in the revised manuscript. **(See revised manuscript, Page 14).**

The discussion is as follows: “It is noteworthy to mention that sample preparation can be optimized through the use of molds specifically designed to produce multiple samples simultaneously. Alternatively, large aerogel samples could be produced and subsequently cut into smaller sizes as needed. To ensure the

structural integrity of the material and prevent damage to the edges during cutting, frozen samples would be utilized. Another approach would be the preparation of large-sized aerogel materials that could be compressed and packaged within a dissolvable net bag. This technique would allow for the direct application of the compressed material into a wound cavity. Upon administration, the net bag would dissolve immediately, enabling the material to expand to its original shape and effectively fill the wound site. This method offers a practical advantage by eliminating the need for small pellets, streamlining the preparation process, and enhancing ease of use in clinical and field settings.”

6. *The author performed hand compression during hemostasis, but the strength of the compression was not given in the article. The addition of this data is conducive to the comparative evaluation of hemostasis effect for readers.*

Response: Currently, there is no standardized or published methodology to precisely calibrate hand pressure in a model like this. However, to address this critique, we have revised the animal study section and clarified our response to this question (**See revised manuscript, Page 33**).

The revision is as follows: “Following treatment, a new set of surgical gauze was employed for a 3-min compression, during which the gauze absorbed blood. The manual pressure was applied with adequate force to stop the bleeding. Specifically, for this groin injury model, pressure was applied using both hands. The right hand applied direct and deep pressure on the hemostatic material, while the left hand pushed on top of the right hand to reinforce the force. Although there is no standardized or published methodology to precisely calibrate hand pressure in a model like this, the combined force was likely in the range of 15–20 lb based on the estimation of our surgeons.”

7. *The hemostatic efficiency of the product Xstat used in the study was not good, and no complete hemostasis was achieved in the five repeats. However, the hemostatic efficiency of this material in other studies is much higher than 80 percent, how do the authors explain this phenomenon?*

Response: We notified that previous studies have reported a hemostatic efficiency of approximately 80% for XStat®. We believe the apparent differences in results arise from variations in how hemostasis was defined and evaluated. In our study, we categorized hemostasis into distinct stages for a more detailed analysis: *i) Immediate hemostasis:* Defined as the cessation of bleeding immediately after treatment (within seconds). XStat® did not achieve immediate hemostasis in our experiments; *ii) Initial hemostasis:* Defined as bleeding cessation following treatment and an additional 3 min of manual compression. XStat® did not achieve initial hemostasis either; *iii) Delayed hemostasis:* Defined as hemostasis achieved at any point during the 3-h observation period. XStat® achieved delayed hemostasis in some cases; and *iv) Stable hemostasis:* Defined as sustained hemostasis without rebleeding throughout the observation period. XStat® failed to achieve stable hemostasis, as rebleeding occurred in multiple instances after delayed hemostasis. In contrast, other studies may have reported an overall success rate of hemostasis (likely equivalent to our definition of delayed hemostasis) without distinguishing between these stages or accounting for rebleeding. In our study, XStat® failed to achieve immediate or initial hemostasis, and its delayed hemostasis was compromised by rebleeding, leading to the death of one out of five subjects in this group. These findings highlight the importance of evaluating not only whether hemostasis is achieved but also how quickly and stably it occurs, particularly in prehospital settings where every second is crucial for survival. (**See revised manuscript, Pages 11-12**).

8. *How persistent is the shape memory performance of the material during long-term storage? With the extension of storage time, will its shape memory effect decrease significantly?*

Response: In our study, we tested the material over one month, and the shape memory properties were consistently retained throughout this duration. Since our materials are stored in dry conditions, the shape memory effect is expected to remain over time under proper storage conditions. The shape memory performance of the material is inherently linked to its dual-scale fiber architecture and material composition. The material is composed of PLA and PCL fibers coated with gelatin. PLA and PCL are widely recognized for their excellent mechanical stability and durability in medical applications. The shape memory effect in this material is primarily a physical process, where the dual-scale fibers are entangled, and the material's porous structure enables significant fluid absorption, facilitating shape recovery. Proper storage conditions are crucial to maintaining the shape memory performance over time. Specifically, a moisture-proof packaging system is recommended to protect the material from environmental factors such as humidity,

which could potentially affect the gelatin coating and fiber interactions. Under these conditions, the shape memory properties of the material are expected to remain stable during long-term storage, with no significant degradation.

9. *One of the advantages of this material is that it can be degraded. However, the whole article did not do any test on degradation, which needs to be added.*

Response: We thank the reviewers for this insightful comment. The materials used in this study—PLA, PCL, and gelatin—are well-established as biodegradable components. Numerous studies have investigated their biodegradation profiles in both preclinical and clinical settings.¹⁻⁶ However, the current animal model we employed is a lethal junctional hemorrhage non-survival model, with a 180-minute observation period following injury and prior to animal sacrifice. Due to the non-survival nature of the model, conducting biodegradation studies was not feasible within this experimental framework. It is important to note that our fabrication technique is not limited to PLA and PCL, which have relatively long degradation profiles of approximately two years. Depending on the application, materials with different degradation profiles could be used. For example, PGLA 90:10 has a degradation time of 2–4 weeks and could be employed in applications requiring faster degradation.

References:

1. Farah et al. *Adv. Drug. Deliv. Rev.* **107**, 367–392 (2016).
2. Matumba et al. *Macromol. Mater. Eng.* DOI: 10.1002/mame.202400056 (2024).
3. Pcheliakov et al. *Biomimetics* **8**, 129 (2023).
4. Bertleff et al. *J. Hand. Surg. Am.* **30**, 513–518 (2005).
5. Wiltfang et al. *J. Craniomaxillofac. Surg.* **27**, 207–210 (1999).
6. Sabel and Stummer, *Eur. Spine J.* **13**, S97–S101 (2004).

10. *The author uses PLA to prepare electrospun fibers, but uses PCL to prepare microfibers. What is the basis for choosing these two materials? In addition, the screening process for the diameter of the fiber material and the length of the short fiber is not given. These parameters have a significant effect on the mechanical properties and hemostatic properties of the material. How does the author take this into account?*

Response: PLA is known for its high elastic modulus and rigidity.¹ In the context of aerogels, PLA nanofibers play a key role in enhancing the structural integrity and mechanical stability of the composite, particularly in supporting shape retention and rigidity under mechanical stress. PCL is more flexible than PLA.² The incorporation of PCL microfibers in the aerogel enhances the material's mechanical flexibility and resilience.³ Additionally, PCL's unique properties enable rapid recovery and shape retention, which are crucial in forming a tamponade at the wound site upon contact with blood.⁴ Together, PLA and PCL balance rigidity and flexibility as well as increase their elongation and impact toughness.^{5,6} When combined with gelatin, they contribute to the aerogel's overall performance. This information has been given in the Introduction section. (**See revised manuscript, Page 4**).

This study builds upon our previously developed materials, where we extensively investigated the effects of dual-scale fiber combinations on material properties.³ In the current study, we focused on exploring the effectiveness of these materials as injectable hemostats. During the material development process, we observed that the short length of nano-sized fibers is critical for achieving the reported knotty and entangled structure with micro-sized fibers. Specifically, the average diameter and length of the nanofibers are $0.29 \pm 0.08 \mu\text{m}$ and $96.49 \pm 24.12 \mu\text{m}$, respectively, while the short microfibers have average diameters and lengths of $0.016 \pm 0.0069 \text{ mm}$ and $2.72 \pm 1.13 \text{ mm}$, respectively. (**See revised manuscript, Supplementary Fig. 1a and Fig. 1b**) We also observed that with increasing microfiber diameter, the pore size and mechanical properties increase. However, all these parameters were optimized in our previous studies to achieve superior mechanical properties, cell infiltration, and a bi-continuous entangled network structure.³

References:

1. Farah et al. *Adv. Drug. Deliv. Rev.* **107**, 367–392 (2016).
2. Shahverdi et al. *Sci. Rep.* **12**, 19935 (2022).
3. Shahriar et al. *Nat. Commun.* **15**, 1080 (2024).
4. Chen et al. *Biomaterials* **179**, 46–59 (2018).
5. Simoes et al. *J. Appl. Polym. Sci.* **112**, 345–352 (2009).

6. Matta et al. *Procedia Mater. Sci.* **6**, 1266–1270 (2014).

11. Although the material can be degraded, it is widely used in hemostasis and still needs to be removed later. Therefore, the tensile strength of the material can significantly affect its removal performance. The authors need to give the tensile strength test of the material and compare it with the Xstat material. In addition, under the premise of heavy use, how can you ensure that every aerogel in the wound is completely removed, thereby reducing its adverse effect on wound healing?

Response: Thanks for the insightful comment. We have performed tensile strength tests comparing NMA and XStat® after blood absorption. The results have been included in the revised Supplementary Information. (See revised manuscript, Supplementary Fig. 27)

Supplementary Fig. 27. Tensile strength of NMA and XStat® materials following blood absorption. Data were analyzed using an unpaired Student's t-test for statistical significance (ns = $p > 0.05$), with results presented as mean \pm standard deviation ($n = 3$).

In addition, post-treatment observations of the NMA aerogels are presented in **Supplementary Fig. 26a and 26b**, which demonstrate that the aerogels maintained their structural integrity after 3 h and were easily removable from the wound cavity without further breakdown. This ease of removal can be attributed to the comparable tensile strength of NMA to XStat® and the stabilization achieved after blood clot formation. For XStat®, surgeons typically remove all material during surgery due to its non-biodegradability. In contrast, NMA aerogels, composed of biocompatible and biodegradable materials, provide an advantage: the residual material remaining in the wound may support clot stability and reduce the risk of secondary bleeding during surgery. Moreover, this method can be adapted by utilizing materials with different degradation profiles as needed. Furthermore, to ensure the complete removal, when necessary, the aerogels could also be packaged in small, net pouches similar to the XStat® device, providing an additional strategy to address concerns about material retrieval from the wound cavity.

12. What is the shape memory and shape recovery mechanism of the material?

Response: Thanks for raising this important point. To understand the mechanism of shape memory and shape recovery of the materials, we conducted detailed structural analyses reported in our previous work.¹ Upon applying different compressive strains, the absorbed fluids in the materials were squeezed out, resulting in a compact and dense micro- and nano-fibrillar structure. The GA-crosslinked gelatin and the entanglements between fibers create networks that effectively bridge the fibrillar junctions. These structural features enable stress transmission throughout the materials, ensuring the fibrillar network does not collapse during deformation. At this stage, the compressed materials store a significant amount of elastic potential energy without force dissipation. When this compressed pellet is rehydrated (i.e., brought into contact with fluids), it rapidly absorbs fluids, regains its original shape, and exerts force due to the release of stored elastic potential energy. Importantly, the shape-recovered aerogels demonstrate microstructures nearly identical to their pre-compression state. (See revised manuscript, Fig. 4I) This unique mechanism, driven by the structural resilience of the fibrillar network and the release of elastic potential energy, ensures robust shape memory and recovery properties of our material.

Reference:

1. Shahriar et al. *Nat. Commun.* **15**, 1080 (2024).

13. Biosafety evaluation is an important prerequisite for material application. The biocompatibility of the

material, such as cell viability and inflammatory response during implantation in vivo, was not tested in this article, which need to be supplemented.

Response: Thanks for the valuable suggestion. The fibers in this study were fabricated using PLA, PCL, and gelatin, which are well-known for their biocompatibility and extensive use in biomedical applications. Gelatin, for instance, is the primary component of widely used FDA-approved hemostatic agents such as Surgifoam[®].¹ Similarly, PLA and PCL are key materials in various FDA-approved implantable medical devices due to their excellent safety profiles and biodegradability. Examples of such devices include bone fixation screws like Lactosorb[®],² which is PLA-based, as well as scaffolds for tissue engineering and drug delivery systems such as Neurolac[®],³ and resorbable sutures like Glycolon[™] from Resorba[®],⁴ which are PCL-based. These examples further demonstrate the established biosafety of the materials used in our study, ensuring their suitability for biomedical applications. We also highlighted in the manuscript that the materials were sterilized before application, ensuring biosafety during the study. While we appreciate the importance of biosafety and biocompatibility evaluation, performing such studies was not feasible in our current experimental setup. The animal model employed in this study is non-survival and designed to simulate the junctional hemorrhage. Following treatment, we observed the animals for 3 h and then sacrificed them according to protocol. Since this model involved extensive surgical interventions such as laparotomy, splenectomy, and cystostomy, along with the complete transection of the right femoral artery and vein, subcutaneous implantation studies could not be performed in this model considering the long-term degradation profiles of PLA and PCL. However, based on the extensive literature and previous work by our group and others, PLA, PCL, and gelatin-based fibers have been shown to exhibit excellent biocompatibility. Our prior studies using similar materials in subcutaneous implantation models in rats demonstrated no adverse inflammatory responses and confirmed their biosafety.⁵ These findings align with the consensus from other groups, further supporting the safety profile of the materials used in our study.^{6,7} We believe this addresses the biosafety concerns, and we will consider including additional biosafety studies in future work.

References:

1. Sabel and Stummer, *Eur. Spine J.* **13**, S97–S101 (2004).
2. Wiltfang et al. *J. Craniomaxillofac. Surg.* **27**, 207–210 (1999).
3. Bertleff et al. *J. Hand. Surg. Am.* **30**, 513–518 (2005).
4. Pcheliakov et al. *Biomimetics* **8**, 129 (2023).
5. Shahriar et al. *Nat. Commun.* **15**, 1080 (2024).
6. Narayanan et al. *Adv. Drug. Deliv. Rev.* **107**, 247–276 (2016).
7. Matumba et al. *Macromol. Mater. Eng.* DOI: 10.1002/mame.202400056 (2024).

14. *Some important parameters in the process of materials' preparation are still relatively lacking, such as the molecular weight of PLA and PCL, and the cutting length of short fibers. Complete and clear material preparation parameters are conducive to readers' reference.*

Response: The molecular weights of PLA and PCL are 60 kDa and 80 kDa, respectively. Additionally, the fibers were cryo-cut to lengths of 60, 30, and 20 mm, dispersed in water to form suspensions, freeze-dried, and stored at 4 °C for further use. **(See revised manuscript, Pages 17-18)**

Reviewer #2 (Remarks to the Author):

1. In Fig1 f, the figure are not in the same scale bar (NA, MA and NMA group).

Response: We have carefully reviewed Fig. 1f and made the necessary corrections to ensure that all SEM images (NA, MA, and NMA groups) are consistently presented with the same scale bar. (See revised manuscript, Fig. 1f)

2. In Fig2 all the measurement need see the model and the real membrane.

Response: In Fig. 2, we have presented the mechanical properties of all experimental groups, as indicated in Fig. 2a-c and 2e-i. Fig. 2d provides a schematic representation of the model used for evaluating mechanical properties, and we confirm that all groups followed a similar experimental setup as depicted in Fig. 2d. To further clarify, we have included photographic images of the experiments in Supplementary Fig. 4, 6, and 8. We hope this addition provides further insight into the experimental design and measurements. (See revised manuscript, Supplementary Fig. 4, 6, and 8).

Supplementary Fig. 4| Photographs illustrating the compression and recovery of XStat®, NA, MA, and NMA materials after absorbing human blood at 90% compressive strain. The red dashed lines, with a diameter corresponding to the initial material length, highlight the extent of recovery loss after compression. NA: nanofiber aerogel. MA: microfiber aerogel. NMA: nano- and microfiber aerogels.

Supplementary Fig. 6] a-d, Photographs showing the length of XStat®(a), NA (b), MA (c), and NMA (d) materials before and after 5 cycles of compression and relaxation at 70%, 80%, and 90% compressive strain following absorption of human blood. The red dashed lines, with a diameter matching the initial material length, indicate the recovery loss observed after cyclic compression. NA: nanofiber aerogel; MA: microfiber aerogel; NMA: nano- and microfiber aerogels.

Supplementary Fig. 8] a,b, Photographs showing the length of NMA materials prepared with 6 h (a) and 18 h (b) crosslinking, before and after the 1st, 10th, 25th, 50th, 75th, and 100th cycles of compression and relaxation at 90% compressive strain following human blood absorption. The red dashed lines, with a diameter corresponding to the initial material length, highlight the recovery loss after cyclic compression. NMA: nano- and microfiber aerogels.

3. *Fig3. Need put the real membrane shape and change in the Fig.*

Response: We have included photographic images of the experiments in **Fig. 3a-c.** (See **Supplementary Fig. 9**) We would like to further clarify that **Fig. 3d-j** demonstrates a computational simulation study, and

the models used were developed using finite element analysis (FEA) and computational fluid dynamics (CFD) based on the structural characteristics of materials. These simulations were performed to understand the stress distribution, energy absorption, and blood flow dynamics within the aerogels under different conditions. Since the images in **Fig. 3d-j** are derived from simulation data, it is not feasible to show original shapes in this context. The simulation results are visual representations of the materials' underlying structural and functional characteristics.

Supplementary Fig. 9] a,b, Photographs showing the length of XStat®, NA, MA, and NMA materials before and after compression and relaxation during testing for toughness (a) as well as elastic energy and resilience (b), following the absorption of human blood. The red dashed lines, with a diameter matching the initial material length, indicate the recovery loss observed after cyclic compression. NA: nanofiber aerogel; MA: microfiber aerogel; NMA: nano- and microfiber aerogels.

4. In Fig4 m, the figure is not in the real shape,

Response: Fig. 4m presents cross-sectional SEM images taken after blood absorption to visualize the distribution of blood cells within the materials. These images were captured to provide insight into the microstructural behavior of the aerogels after interacting with blood. It is important to note that the real photographic images of the aerogels cannot show the detailed distribution of blood cells on the surface or within the material's cross-sections. For example, in **Supplementary Fig. 10**, we can observe the overall red color of the aerogels resulting from blood absorption but cannot discern how blood cells are distributed throughout the material. Therefore, SEM imaging was employed as an essential technique to achieve this level of detail.

We hope this explanation clarifies why SEM imaging was utilized and why the real photographic images cannot depict the intricate details shown in Fig. 4m.

5. Fig5 a. why the is no different in the NA group, MA group and NMA group? And why the clotting time is different in NA group, MA group and NMA group? The Fig5 h need to put in the position of Fig5a.

Response: Fig. 5a illustrates the blood clotting index, and no significant differences were observed among the NA, MA, and NMA groups. Similarly, Fig. 5b shows no statistically significant differences in clotting time across these groups. The statistical analyses were conducted using one-way ANOVA with Tukey’s multiple comparisons test in both cases. Although the NMA group shows a slightly lower blood clotting time, this difference is not statistically significant. This slight reduction could be attributed to the rapid blood absorption properties of the NMA group, as demonstrated in Fig. 4.

We appreciate the reviewer’s suggestion regarding the rearrangement of Fig. 5. However, Fig. 5h is a schematic diagram illustrating the pro-coagulant mechanism of NMA. In Fig. 5a and 5b, we have presented the blood clotting efficiency of the materials, followed by Fig. 5c, 5d, and 5e, where we demonstrated the reasons behind this efficiency—specifically the absorption of RBCs and platelets. Furthermore, we have shown the activation of the blood coagulation pathway by the aerogels in comparison to XStat® in Fig. 5f and 5g. Finally, Fig. 5h serves as a schematic representation to summarize and explain the action mechanism of NMA. We believe this arrangement is well-organized and logical: first, we present experimental evidence, and then we provide a visual representation of the mechanism. We feel that keeping Fig. 5h in its current position, at the conclusion of the experimental data, is appropriate as it ties all the findings together. Therefore, we have decided not to move Fig. 5h and hope the reviewer will appreciate this logical flow of the presentation

6. in line 217-220, why the author did not choose the same long and size of the Xstat and NMA materials.

Response: For visual comparison purposes, we used a 5.5 cm long NMA and a 3 cm long XStat®. The reason for this was to demonstrate the rapid shape recovery properties of NMA materials. Specifically, a 5.5 cm long NMA fully recovered its shape within 5 s upon contact with blood, while a 3 cm long XStat® recovered only 0.4 cm within the same period. (See revised manuscript, Supplementary Fig. 12). This striking visual difference, as shown in Supplementary Movies 8 and 9, highlights the superior performance of NMA. Moreover, to ensure proper quantification and avoid any bias caused by the length difference, we presented the data in a "time per length" format. (See revised manuscript, Fig. 4k and Supplementary Table 9). Fig. 4k shows how much time is required for a material to recover per centimeter of its length. By normalizing the recovery time to length, we ensured a fair and easy comparison between our materials and XStat®, as well as others, regardless of the actual material lengths used in the visual demonstrations.

7. In line 379-381, “The aerogel demonstrated 380 excellent mechanical properties, including a high specific elastic modulus, high resilience, rapid shape recovery, and exceptional robustness”, the author need to put the real comparison data of the high specific elastic modulus, high resilience, and rapid shape recovery with other group.

Response: To the best of our knowledge, there is limited published data specifically comparing the specific elastic modulus and resilience of shape memory hemostats. For a comparison of shape recovery data with other hemostats, we have provided clearer comparisons in Supplementary Table 9.

Supplementary Table 9. Comparison of the shape recovery rate of NMA in blood with all other hemostats reported in the literature known for quick shape-recovery properties in blood.

No.	Hemostats Under Development	Shape Recovery Rate in Blood (s/cm)		References
		Mean	s.d.	
1	NMA	0.90	0.14	This Work
2	MA	1.09	0.14	
3	NA	1.21	0.31	
4	XStat®	7.27	0.90	
5	Nanofiber Peanut (NFP)	2.63	0.32	1
6	MACS1	4.00	0.81	2

7	MACS2	2.50	0.40	
8	QCS/PDA0.5	72.40	11.10	
9	QCS/PDA1	47.60	6.90	
10	QCS/PDA2	41.60	7.00	3
11	QCS/PDA3	31.00	4.90	
12	QCS/PDA4	19.80	4.90	
13	GT25/DA0	26.20	1.10	
14	GT25/DA2	23.55	1.60	
15	GT25/DA4	21.20	1.15	4
16	GT25/DA6	12.10	0.85	
17	GT25/DA8	11.70	1.25	
18	GT25/DA10	12.00	0.80	
19	GCS	10.00	-	
20	GCSF-1	10.00	-	
21	GCSF-2	6.000	-	5
22	GCSF-3	5.000	-	
23	GCSF/CT	5.000	-	
24	CS20/PDA1.5	8.667	1.13	6
25	CS20/PDA3	7.533	1.00	
26	CS20/PDA4.5	6.20	0.80	6
27	CS20/PDA6	5.13	0.40	
28	CS20/PDA7.5	4.47	0.80	
29	QCSG/CNT0	7.70	-	7
30	QCSG/CNT4	16.15	-	
31	CA1, 160 °C, 12 h	41.10	2.70	
32	CA2, 180 °C, 10 h	27.04	1.40	8
33	GS	306.50	18.32	
34	A-spCS	4.37	1.36	9
35	spCS	4.00	1.13	

NA: nanofiber aerogels; MA: microfiber aerogels; NMA: nano- and microfiber aerogel; MACS: microchannelled alkylated chitosan sponge; QCS/PDA: quarternized chitosan/polydopamine; GT/DA: gelatin/dopamine; GCS: glutaraldehyde/chitosan; GCSF: glutaraldehyde/chitosan/silk fibronin short nanofibers; GCSF/CT: glutaraldehyde/chitosan/silk fibronin short nanofibers with built in cargo-loaded CaCO₃ and protonated tranexamic acid; CS/PDA: chitosan/polydopamine; QCSG/CNT: glycidyl methacrylate-functionalized quarternized chitosan/carbon nanotube; CA: carbonized gelatin aerogel; GS: gelatin sponge; A-spCS: alkylated superporous chitosan sponge; spCS: superporous chitosan sponge.

References:

1. Chen et al. *Biomaterials* **179**, 46–59 (2018).
2. Du et al. *Nat. Commun.* **12**, 4733 (2021).
3. Li et al. *ACS Appl. Mater. Interfaces* **12**, 35856–35872 (2020).

4. Huang et al. *Chem. Mater.* **32**, 6595–6610 (2020).
5. Lu et al. *Research* **6**, 0150 (2023).
6. Zhao et al. *Chem. Eng. J.* **403**, 126329 (2021).
7. Zhao et al. *Nat. Commun.* **9**, 2784 (2018).
8. Yang et al. *Adv. Sci.* **10**, e2207347 (2023).
9. Jiang et al. *Nat. Commun.* **15**, 5460 (2024).

8. *In line 411-419, the hemostasis mechanism is complicate, it needs blood coagulation factors, not just RBCs and platelets accumulate, pls put more evidence for the coagulation procedure of NMA materials.*

Response: In lines 411-419 of the original manuscript, we proposed a hemostatic mechanism involving the interaction of blood components with NMA materials. To clarify, blood coagulation is indeed a complex process that involves the interaction of platelets, erythrocytes, and coagulation proteins.¹⁻⁵ The NMA materials not only accumulate RBCs and platelets but also interact with coagulation factors present in the blood. The NMA aerogels absorb blood from the wound cavity, which inherently includes coagulation factors. This absorption facilitates their concentration within the nanofiber network, creating a localized environment conducive to coagulation. Additionally, the gelatin component in the materials synergistically activates both intrinsic and extrinsic coagulation pathways. This activation enhances the aggregation and activation of platelets and RBCs, leading to the formation of a stable clot and achieving hemostasis. We emphasize that the NMA materials themselves do not require external coagulation factors. Instead, their nanofiber architecture and gelatin content act as a scaffold that captures coagulation factors and supports their activation. These processes ensure effective hemostasis through intrinsic and extrinsic pathways. To support this mechanism, we have provided additional evidence, which detail the interaction of NMA with blood components and the pathways involved. The data show how the materials trap and concentrate coagulation factors, ultimately stabilizing the clot and preventing rebleeding. (**See revised manuscript, Fig. 4 and 5, and Supplementary Tables 20–24**)

References:

1. Du et al. *Nat. Commun.* **12**, 4733 (2021).
2. Zhao et al. *Nat. Commun.* **9**, 2784 (2018).
3. Jiang et al. *Nat. Commun.* **15**, 5460 (2024).
4. Kibria et al. *Adv. Mater.* **30**, 49 (2018).
5. Andrabi et al. *Bioact. Mater.* **38**, 154-168 (2024).

9. *In line 431-432, the author use the recent report data, the condition is changed, the author need to do the experiment in the same condition, so the data of control, XStat®, and QCG® groups from the recent report need to make sure.*

Response: We would like to clarify that all experiments were conducted under the same protocol approved by the Institutional Animal Care and Use Committee (IACUC) at the University of Nebraska Medical Center (UNMC). While the data from the control, XStat®, and QCG® groups were published earlier, the experiments were performed under the identical conditions and within the same time frame as the current study. Due to the nature of the animal study, each pig required almost an entire day for preparation, treatment, and observation. Therefore, the experiments were conducted individually on each pig. Nevertheless, consistent conditions were maintained across all groups, including the same surgeon, surgical procedures, and environmental settings. Moreover, in alignment with the "3Rs" principle of animal research (Replacement, Reduction, and Refinement), the Comparative Medicine team at UNMC overseeing our animal protocol did not permit repeating experiments for comparison purposes unless new findings or significant differences were anticipated. Since the earlier data from the control, XStat®, and QCG® groups align with our current findings under the same protocol, there was no scientific or ethical justification for repeating the experiments. We have also acknowledged this as a limitation of the study in the Discussion section, emphasizing that while all groups were conducted under consistent conditions, the inability to conduct simultaneous experiments with all groups due to logistical constraints could be perceived as a limitation. (**See revised manuscript, Pages 16-17**)

Reviewer #3 (Remarks to the Author):

In this manuscript, the author developed bicomponent aerogels based on poly (lactic acid) (PLA) nanofibers and poly(ϵ -caprolactone) (PCL) microfibers for effective halting bleeding on junctional hemorrhage. The aerogel exhibited excellent absorption capacity and in vitro blood clotting efficacy. This work is interesting, well-structured, and logical. Therefore, I recommend it has the potential for publication in Nature Communications after major revisions.

Response: Thanks for the reviewer's positive and encouraging feedback. We are delighted that the reviewer finds our study interesting, well-structured, and logical. We greatly appreciate the recognition of the potential impact of our findings. We have carefully addressed the points raised and revised the manuscript accordingly.

Some specific comments:

1. The whole manuscript has no line and page numbers.

Response: Page numbers have been added to the revised manuscript. The line numbers will be automatically generated in the PDF version by the journal's submission system.

2. The first three sentences of the Introduction section cite a total of 20 references, and I think this is an overquoting phenomenon.

Response: We understand that citing 20 references within the first three sentences might appear excessive. We have carefully reviewed the cited references and reduced the number to only the most relevant and impactful ones that support the statements. **(See revised manuscript, Page 3).**

3. The abbreviations of NMA need to be defined in the first appearance.

Response: The abbreviation "NMA" has been appropriately defined in both the abstract and the introduction in its first appearance. **(See revised manuscript, Pages 2-3).**

The change in the Abstract is as follows: "Here, we report a bicomponent nano- and microfiber aerogel (NMA) with super-elastic properties and enhanced resilience that can be injected into deep, narrow junctional wounds, effectively halting bleeding."

The change in the Introduction is as follows: "To overcome the abovementioned problems, we report a bicomponent nano- and microfiber aerogel (NMA) primarily composed of polylactic acid (PLA) nanofibers (NFs) and poly(ϵ -caprolactone) (PCL) microfibers (MFs) for the management of JH."

4. "Due to its high absorption properties, the PLA NF component could effectively entrap platelets and clotting factors and enhance the aggregation of red blood cells (RBCs)." "Due to its high absorption properties" is repeated twice.

Response: We have revised the sentence to avoid repetition. **(See revised manuscript, Page 4).**

The change is as follows: "Due to its high absorption properties, PLA NF is intended to entrap platelets and clotting factors, enhancing the aggregation of red blood cells (RBCs)."

5. Are there long-term follow-up studies to evaluate the hemostatic efficacy and safety of aerogels, including their biodegradation process, long-term safety, and potential impacts on surrounding tissues?

Response: The purpose of this study was to develop and evaluate aerogels for prehospital management of junctional hemorrhage. As such, the study focused on their ability to control bleeding for a few hours, ensuring the patients survive long enough to reach surgical care. Evaluating long-term hemostatic efficacy was not feasible in the current non-survival animal model, as the goal was to simulate emergency scenarios rather than prolonged usage. Nevertheless, we acknowledge the importance of long-term hemostatic performance, which could be explored in future studies with appropriate survival models.

In terms of safety and biodegradation, the fibers used in our aerogels are composed of PLA, PCL, and gelatin, which are widely recognized for their biocompatibility and extensive use in FDA-approved medical devices such as sutures, bone fixation screws, and hemostatic agents. These materials have well-documented biodegradation profiles, with PLA and PCL degrading into non-toxic byproducts and gelatin

being enzymatically degraded in the body.¹⁻⁴ Although this study did not investigate their long-term degradation behavior due to the experimental model's constraints, our prior work and literature confirm their biosafety and controlled biodegradability in biomedical applications.^{5,6}

Regarding the potential impact on surrounding tissues, the aerogels were designed with tunable mechanical properties to maintain flexibility and avoid excessive pressure on wound walls. As shown in **Supplementary Fig. 26**, the aerogels can be easily removed from wounds without causing additional damage, and the materials conform to wound geometry without exerting harmful pressure. Furthermore, prior subcutaneous implantation studies with similar materials demonstrated no adverse inflammatory responses, suggesting their excellent compatibility with surrounding tissues.⁷ We appreciate the reviewer's suggestion and will evaluate additional biosafety and long-term efficacy in future work.

References:

1. Sabel and Stummer, *Eur. Spine J.* **13**, S97–S101 (2004).
2. Wiltfang et al. *J. Craniomaxillofac. Surg.* **27**, 207–210 (1999).
3. Bertleff et al. *J. Hand. Surg. Am.* **30**, 513–518 (2005).
4. Pcheliakov et al. *Biomimetics* **8**, 129 (2023).
5. Narayanan et al. *Adv. Drug. Deliv. Rev.* **107**, 247–276 (2016).
6. Matumba et al. *Macromol. Mater. Eng.* DOI: 10.1002/mame.202400056 (2024).
7. Shahriar et al. *Nat. Commun.* **15**, 1080 (2024).

6. *Although the article mentions that the material could potentially be used in battlefield or emergency rescue situations, not provide any experimental data or discussion regarding the material's performance under harsh environmental conditions, such as extreme temperatures, humidity, or pressure.*

Response: We sincerely thank the reviewer for this insightful comment. As this is a proof-of-concept study, our primary objective was to evaluate the hemostatic efficacy and biocompatibility of the hybrid aerogels. While we did not directly assess the material's performance under extreme environmental conditions such as temperature, humidity, or pressure, we believe that a robust packaging system could mitigate these potential challenges. Since the aerogels are composed solely of polymer-based materials and do not rely on biologically active compounds, proper packaging could preserve their structural and functional integrity in harsh conditions. However, we have acknowledged this limitation and highlighted the need for future studies to systematically test the material under such conditions. **(See revised manuscript, Page 17).**

The revision is as follows: “Also, we did not assess the material's stability under extreme environmental conditions, such as high temperatures, humidity, or pressure, which are critical for battlefield or emergency applications. Future studies are necessary to address these conditions and validate the aerogels' functionality in such scenarios.”

7. *The study conducted comparative experiments between the aerogel and XStat®, demonstrating the aerogel's advantages in shape memory and absorption capacity. However, the specific differences in the hemostatic mechanisms between the aerogel and traditional hemostatic materials were not mentioned.*

Response: We thank the reviewer for highlighting the importance of comparing the hemostatic mechanisms of XStat® and QuikClot Combat Gauze (QCG®) with NMA. In response, we have added a detailed comparison as follows to the Discussion section, explaining how XStat® and QCG® promote hemostasis but may be insufficient for severe junctional hemorrhages. We also highlighted the unique advantages of our hybrid aerogels, which address these challenges through a combined mechanical and hemostatic approach. **(See revised manuscript, Pages 15-16).**

The discussion is as follows: “We propose the hemostatic mechanism of NMA in JH as follows: (i) compressed dry NMA store elastic energy due to efficient stress transmission by the bi-network; (ii) upon injection into a bleeding cavity, the porous fiber architecture rapidly absorbs blood, allowing the aerogel to return to its original shape; (iii) the shape recovery effect exerts pressure on the wound, creating a physical barrier that quickly stops bleeding; (iii) the intertwined fibrous structure concentrates coagulation factors, enhancing the aggregation of erythrocytes and activated platelets; and (v) RBCs and platelets accumulate on the surface and within aerogels and are activated, achieving stable hemostasis and preventing rebleeding. In contrast, the hemostatic mechanisms of XStat® and QCG® differ significantly from NMA. XStat® employs chitosan granules that promote platelet adhesion and aggregation while interacting with red blood cells and proteins, initiating clot formation. QCG®, on the other hand, relies on kaolin, an inorganic

mineral that activates Factor XII in the coagulation cascade to generate thrombin and form fibrin clots. However, these mechanisms are limited in addressing the complexity of severe JH, as evidenced by their reduced effectiveness in our non-survival uncontrolled model.”

8. *When conducting the aerogel compression test, the author should show the actual image of the aerogel in the test process in Figure 2, so as to show the ability of shape recovery of the aerogel more intuitively.*

Response: We have added photographic images of the aerogel during the compression test to better illustrate its shape recovery ability. Pls refer to the response to Rev#2, Question 2. **(See revised manuscript, Supplementary Fig. 4, 6, and 8)**

9. *The author should briefly describe the chemical properties of PLA and PCL for understanding the role of PLA and PCL in gelatin-based aerogels.*

Response: We appreciate the reviewer’s request for a brief description of the chemical properties of PLA and PCL in aerogels. We have added more details. **(See revised manuscript, Page 4)**

The revision is as follows: “PCL and PLA are biodegradable, biocompatible polyesters with hydrophobic backbones lacking reactive groups.¹ They are primarily synthesized via ring-opening polymerization (ROP) of α -hydroxy acid monomers.”

Reference:

1. Kaluzynski et al. *Macromolecules* **55**, 2210-2221 (2022).

10. *Do aerogels have antibacterial properties? Whether the author consider adding antibacterial agent to the aerogels to prevent infection at the junctional wound site?*

Response: We appreciate the reviewer’s valuable comment regarding the antibacterial properties of the aerogels. The current composition of the aerogels does not inherently possess antimicrobial properties. We could incorporate an antimicrobial agent into the aerogels to prevent infections using different approaches (e.g., mixed with gelatin, absorption). Alternatively, we could replace the gelatin component with chitosan, which imparts antibacterial activity. However, we opted not to include antibacterial agents in the hemostats at this stage due to concerns about increasing production costs, manufacturing complexity, and difficulties in regulatory approval. In addition, our study focuses on the prehospital management of lethal hemorrhage, where the aerogel is intended for use only for a short duration—just long enough to stabilize the patient during transport to the hospital (e.g., 1-3 h). Once the patient arrives, the surgeon can manage the wound and its potential risks, including infection. In a separate project, we are currently exploring the incorporation of antimicrobial peptides into aerogels for treating infectious wounds. If needed, we can submit the preliminary data to facilitate the reviewing process of this work. Thanks for the thoughtful suggestion, which has provided us an opportunity to clarify the intended use and future directions of this research. **(See revised manuscript, Page 16).**

The discussion is as follows: “Biological molecules could be incorporated into NMA through various methods (e.g., coating, conjugation, and encapsulation) to regulate cell behavior or infection management, further enhancing their capability in tissue regeneration. While the current NMA formulation does not include antibacterial properties, incorporating antimicrobial agents, such as chitosan or antimicrobial peptides, could prevent infection at wound sites. However, this study focuses on the prehospital management of lethal hemorrhage, where NMA is intended for short-term use to stabilize patients during transport to the hospital. Given concerns about increased production costs, manufacturing complexity, and regulatory hurdles, antibacterial agents were not included at this stage. Future studies may explore antimicrobial incorporation for extended applications, such as infectious wound treatment, where NMA’s biodegradable nature could offer additional therapeutic benefits. In contrast to non-biodegradable XStat[®], which requires enclosed pellets for surgical removal, NMA’s biodegradability allows portions of the material to remain in the wound, potentially aiding tissue regeneration post-application.”

11. *Will the aerogel remain partially in the wound after use? Authors should determine the water contact angle to investigate wettability of aerogel, and observe the integrity of the aerogels after fully absorbing blood.*

Response: We appreciate the reviewer’s question regarding the aerogel’s presence in the wound after use. Once the patient reaches the hospital, the surgeon removes all the XStat[®] samples during surgery due to

their nonbiodegradability. As aerogels are made of biocompatible and biodegradable materials, it is expected that the aerogels could remain partially in the wound after use, which may reduce the chances of secondary bleeding during surgery. This is one of the potential advantages of aerogels. For this case, we have to optimize their degradation profiles and validate their long-term biosafety and biocompatibility.

We also appreciate the reviewer's insightful comment regarding the water contact angle and the integrity of the aerogel after absorbing blood. PCL and PLA are hydrophobic. We have performed plasma treatment and gelatin-coating to enhance wettability. As noted, the wettability of the aerogels was evaluated using both water and a high-viscosity fluid, human blood. Due to the aerogels' extremely high absorption capacity, the water contact angle could not be measured using standard methods, as the liquid was absorbed almost instantaneously upon contact with the material. This behavior indicates complete wetting, with a contact angle of effectively zero. Similarly, when tested with blood, the aerogels demonstrated rapid absorption, further confirming their zero contact angle and exceptional wetting behavior in high-viscosity fluids. **(See revised manuscript, Page 9 and Supplementary Movie 10)**

To address concerns about the aerogel's structural integrity after absorption, we refer to **Supplementary Movies 7 and 9**, which demonstrate the rapid absorption and recovery process. We also showed post-treatment observations of the NMA aerogels. After 3 h, the aerogels maintained their structure and integrity, as shown in **Supplementary Fig. 26a and 26b**. Post-treatment observations within the wound cavity confirmed that the NMA aerogels effectively filled the wound area and conformed to the wound shape, demonstrating their ability to form a tamponade and maintain their shape despite the pressure of bleeding without being expelled by excessive blood flow. Additionally, the tensile strength of NMA after absorbing blood is comparable with that of XStat[®]. **(See revised manuscript, Supplementary Fig. 27)**

Supplementary Fig. 27 | Tensile strength of NMA and XStat[®] materials following blood absorption. Data were analyzed using an unpaired Student's t-test for statistical significance (ns = $p > 0.05$), with results presented as mean \pm standard deviation ($n = 3$).

12. "The compressed NMA pellets could be injected into the deep and narrow wounds, where they might re-expand to form a tamponade and stop the bleeding". The manuscript stated that NMA pellets are injected into wounds, but the Supplementary Materials stated that NMA pellets are submerged in wounds. Is it injection or submersion? The compressed NMA pellets Size?

Response: The NMA pellets are injected into the wound, as demonstrated in **Supplementary Movie 20**. We have corrected this in the Supplementary Materials. The volume of compressed NMA pellets is $\sim 200 \text{ mm}^3$. The fixed and recovered sizes for each shape-memory hemostat have been summarized in **Supplementary Table 7**. **(See revised manuscript, Supplementary Information Page S67)**

13. "Additionally, pore diameter analysis revealed values of $236.70 \pm 58.03 \mu\text{m}$ and $183.10 \pm 48.98 \mu\text{m}$ for MA and HA", what is HA?

Response: "HA" should be "NMA". We have corrected this. **(See revised manuscript, Page 5)**

14. The format of the references needs to be checked carefully.

Response: We have reformatted the references carefully. **(See revised manuscript, Pages 37-42).**

15. *The manuscript requires English revision. There are detailed mistakes throughout the manuscript that need to be corrected.*

Response: We have thoroughly reviewed and refined the language in the revised manuscript.

Reviewer #4 (Remarks to the Author):

1. What are the noteworthy results?

Reviewer's Evaluation: NMA is superior to QCG and Xstat (established hemostatics) in junctional hemorrhage.

Response: Thanks for the positive comment. We appreciate the recognition of the results showing NMA's superiority over QuikClot® Combat Gauze (QCG) and XStat® in managing junctional hemorrhage.

2. Will the work be of significance to the field and related fields? How does it compare to the established literature? If the work is not original, please provide relevant references.

Reviewer's Evaluation: The authors describe a novel biogel, that has hemostatic properties, and improved survival and hemostasis in a porcine model of uncontrolled femoral hemorrhage. Overall, the study is interesting, and novel materials to produce hemostasis from uncontrolled junctional hemorrhage are needed. The work may be of benefit to civilian and military patients with junctional haemorrhage.

Response: Thanks for the thoughtful comments and evaluation. We appreciate the recognition of the novelty and potential significance of our study, particularly in the context of hemostatic materials for managing uncontrolled junctional hemorrhage.

3. Does the work support the conclusions and claims, or is additional evidence needed?

Reviewer's Evaluation: There are some shortcomings that need to be discussed, below.

Response: Thanks for valuable feedback. We have carefully considered each comment and have provided additional explanations or modifications to the manuscript where necessary. We believe the revised version now includes the additional evidence and clarifications required to strengthen the conclusions and claims. We hope that these changes adequately address the concerns and that the updated manuscript meets the reviewer's expectations.

4. Are there any flaws in the data analysis, interpretation and conclusions? Do these prohibit publication or require revision?

Reviewer's Evaluation: There is no clearly stated hypothesis and research question to be answered. Even though there seems to be answers to a question. Please state this more clearly in the introduction.

Response: We have revised the last paragraph of the Introduction section to ensure the hypothesis and research questions are explicitly stated. (See revised manuscript, Pages 3-4).

The revision is as follows: "We hypothesize that the unique structural design of the aerogel integrates complementary properties, enabling tunable mechanical properties, rapid shape recovery in blood, absorption and coagulation activation, and providing a tamponade effect, thereby ensuring effective management of lethal junctional hemorrhages."

5. The statistical analyses are sound. One- and two-way ANOVA with Tukey's multiple comparisons test seem correct analyses for the data (depending on distribution). Survival analysis using log-rank and Fisher also seem correct for the categorical data. Information on data distribution is missing. Did you perform Smirnov-Kolmogorov or Shapiro-Wilks to test for normality? Statistical analyses are only provided in the figure legends. It would be good to see a dedicated "statistical analysis" section in the method part. The graphical work is clear and readable.

Response: We appreciate the reviewer's positive feedback on the clarity and soundness of our statistical analyses and graphical representations. We acknowledge the reviewer's concern regarding the missing information about data distribution and normality testing. To address this, we performed the Shapiro-Wilk test to assess data normality, as it is suitable for our sample size. This additional information has been included in the revised manuscript. (See revised manuscript, Page 37).

We agree that a dedicated "Statistical Analysis" section would enhance the clarity and accessibility. We initially included this section in the Supplementary Materials, but it has now been moved to the main manuscript for improved accessibility. (See revised manuscript, Page 37).

The revision is as follows: "**Statistical analysis.** The Shapiro-Wilk test was used to assess data normality, and the data were presented as mean values with corresponding standard deviations (s.d.). Statistical analyses were conducted utilizing analysis of variance (ANOVA) with GraphPad Prism (version 9.5.1).

Pairwise comparisons were carried out using either ordinary one-way or two-way ANOVAs, followed by Tukey's multiple comparisons post hoc test when appropriate. Additional statistical comparisons were performed as indicated. For measurements obtained from photographs or SEM images, Image J was utilized after calibrating pixels to millimeters or micrometers. Survival analysis was performed using the log-rank tests to compare survival curves among groups. The Fisher exact test was employed for categorical data comparisons to assess statistical significance. All analyses were conducted with a significance level set at 0.05. Graphs and figures were created and visualized using GraphPad Prism, Origin Pro (version 8.5), and BioRender."

6. *Is the methodology sound? Does the work meet the expected standards in your field?*

Reviewer's Evaluation:

6.1. *How did you decide on the number of animals in each group, n=5? Did you perform an a priori power calculation?*

Response: We appreciate the reviewer's feedback regarding the methodology. Regarding the number of animals in each group ($n = 5$), the sample size was determined based on a power analysis aimed at detecting clinically significant differences in survival rates between untreated and treated groups. Specifically, we used a two-sided test for comparing two proportions, and the power analysis indicated that a sample size of 5 animals per group would provide sufficient power (80%) to detect a significant effect with an alpha level of 0.05 and a beta level of 0.2. The power analysis calculation was provided in the supplementary methods section. We have now added this information to the main manuscript. (**See revised manuscript, Page 36**).

6.2. *How was the allocation of treatment elected? Randomly? Non-randomly?*

Response: Regarding randomization, due to the nature of the experiment, randomization was not performed. All animals were treated in the same manner, with identical preparation and observation protocols to ensure consistency. Each pig model required approximately 7–8 h, including preparation and observation periods, and only one pig could be handled per day. These constraints prevented randomization, and we have disclosed this limitation in the ARRIVE guidelines table provided in **Supplementary Tables 15, Pages S90-S93**. We also confirm that the pigs were statistically non-significant among the experimental groups (**Supplementary Fig. 14**). There were no significant differences in baseline measures such as mean arterial pressure (MAP), heart rate (HR), end-tidal carbon dioxide (EtCO₂), body temperatures, hemoglobin levels, platelets, lactate levels, normal prothrombin time (PT), activated partial thromboplastin time (aPTT), international normalized ratio (INR), or fibrinogen levels (**Supplementary Fig. 15-17**). These results demonstrate the comparability of the groups at baseline.

6.3. *Was any part of the study blinded? All should be included according to the ARRIVE guidelines.*

Response: Regarding the blinding of the study, we confirm that the UNMC Comparative Medicine's administrative officer, who recorded baseline vitals such as MAP, HR, EtCO₂, and body temperatures, was blinded to the treatment samples. Additionally, the pathologist who measured hematologic values (such as hemoglobin, hematocrit, platelets, and lactate levels) and coagulation data (including prothrombin time (PT), activated partial thromboplastin time (aPTT), international normalized ratio, fibrinogen levels, and blood pH) was blinded to the treatment samples. (**See revised manuscript, Page 17**) However, as described earlier in the response to randomization, blinding the surgeon performing the procedure was not feasible, as only one pig was handled per day. Given the time-consuming nature of the procedure and the need for expert surgical intervention, the surgeon could not be blinded to the sample. All other details concerning adherence to the ARRIVE guidelines, including the methods for animal allocation, randomization, and blinding, are summarized in **Supplementary Table 15, Pages S90-S93**.

6.4. *What was the average weight of the pigs?*

Response: The average weight of the pigs is shown in **Supplementary Fig. 14**, which indicates no significant difference in body weight among the experimental groups.

6.5. *How were the pigs sedated, medications and doses? How were they monitored?*

Response: The animals were fasted for 12-h before surgery, with free access to water. On the day of surgery, premedication was administered, and an intravenous (IV) line was established in a marginal ear vein using a 20–22-gauge Angiocatheter (Nordson Medical, Loveland, CO 80538, USA). Anesthesia was

induced via isoflurane and oxygen (3-5 L/min) using a veterinary anesthesia ventilator (MDS Matrx 3000, HALLOWELL EMC, Pittsfield, MA 01201, USA) to facilitate intubation. During the procedure, anesthesia was maintained via isoflurane and oxygen (1-2 L/min) until death or euthanasia. Vital signs were continuously monitored, with a rectal temperature probe and EKG (cardiac) monitors (DRE Waveline VS, M16C13140003, PET PRO SUPPLY CO.®, Frisco, TX, USA) in place. The swine were positioned on a water-circulated warming blanket (BLANKETROL® II, 222R, Cincinnati Sub-Zero Products, Inc., Cincinnati, OH 45241, USA), maintained at 103 °F. Mechanical ventilation was set at 12-15 breaths per min with a tidal volume of 5-10 mL/kg (MDS Matrx 3000, HALLOWELL EMC, Pittsfield, MA 01201, USA), and end-tidal pCO₂ was maintained between 35-45 mmHg. All incisions were made using an electrosurgical generator (Valleylab Force 1C, Pfizer, New Bedford, MA, USA). For pressure monitoring and blood sampling, a 20-gauge Angiocatheter was surgically placed in the carotid artery and a 14–16-gauge Angiocatheter was inserted in the jugular vein for fluid and medication administration. These catheters (Nordson Medical, Loveland, CO 80538, USA) were inserted through a surgical cutdown in the right or left neck (**Supplementary Fig. 18**). Following splenectomy, warm lactated Ringers (LR) solution was administered at three times the spleen's weight at a rate of 100 mL/min using a roller infusion pump (DOSE IT P910, Integra Biosciences AG, Tardisstrasse, Zizers 7205, Switzerland). The post-treatment observation period was conducted under general anesthesia. (**See revised manuscript, Pages 30-35**).

6.6. *Where did you measure arterial blood pressure?*

Response: Arterial pressure was monitored by a designated technician from UNMC Comparative Medicine, who was blinded to both samples and subjects at predetermined time intervals during the experiments (e.g., 0 min, 15 min, 30 min, 60 min, 120 min, and 180 min) using an in-house EKG monitor (DRE Waveline VS, M16C13140003, PET PRO SUPPLY CO., Frisco, TX, USA). (**See revised manuscript, Page 34**).

6.7. *A methodology section is missing. The rationale and explanation of the haemostatic agents are extensive and informative. However, the methodology section needs improvement, although the movie material is informative.*

Response: We apologize for any confusion. Previously, the methodology section was included as supplementary information in accordance with *Nature Nanotechnology* guidelines. For the revised manuscript, as per *Nature Communications* rules, we have transferred the methodology section into the main text. The Methodology section provides specific details about the surgical procedure, administration of materials, and monitoring protocol to ensure that the methodology is reproducible and well-defined. (**See revised manuscript, Pages 17-37**)

7. *Is there enough detail provided in the methods for the work to be reproduced?*

Reviewer's Evaluation: It would probably be possible with all the informative supplemental material. The methodology section needs improvement.

Response: We have thoroughly revised the Methods section to improve clarity. Specifically, we have ensured that critical experimental steps, animal preparation protocols, surgical procedures, and analysis methods are explicitly described. Furthermore, we have cross-referenced the relevant Supplementary Materials, where detailed protocols are provided, to facilitate reproducibility. (**See revised manuscript, Pages 17-37**).

8. *Suppl fig 21. Was randomization between treatments performed?*

Response: Due to the nature of the experiment, randomization was not performed. All animals were treated in the same manner, with the identical preparation and observation protocols, ensuring consistency across the study. The experiment requires significant time and expertise, as each pig model takes approximately 7–8 h, including preparation and observation periods, and only one pig can be handled per day. This limitation, along with the rationale for not performing randomization, is disclosed in the ARRIVE guidelines table provided in **Supplementary Tables 15, Pages S90-S93**. Additionally, we confirm that the pigs were statistically non-significant among the experimental groups (**Supplementary Fig. 14**). There were no significant differences in baseline mean arterial pressure (MAP), end-tidal carbon dioxide (EtCO₂), heart rate (HR), body temperatures, hemoglobin levels, platelets, lactate levels, normal prothrombin time (PT), activated partial thromboplastin time (aPTT), international normalized ratio (INR), or fibrinogen levels (**Supplementary Fig. 15-17**). These findings further demonstrate the comparability of the groups at baseline.

9. The authors performed a splenectomy before hemorrhage, and a laparotomy. This is invasive and may confound results in terms of hemostasis, inflammation and more. Please comment on the rationale and limitations. Splenectomy before bleeding is not always necessary (PMID: 26974424).

Response: We appreciate the reviewer's comment regarding the potential confounding effects of splenectomy and laparotomy on hemostasis, inflammation, and other physiological variables. In our study, we employed an uncontrolled hemorrhage model characterized by rapid blood loss, exceeding rates reported in controlled hemorrhage models such as the one described in the referenced study (Boysen et al. *Shock* **46**, 439-46 (2016)). Unlike their model, which used a controlled hemorrhage rate of 3–4 mL/kg/min with an endpoint of 40 mmHg MAP and blood return if MAP dropped below 35 mmHg, our model involves severe blood loss at a significantly higher rate (over 15 times faster) and MAP values falling below 15 mmHg. Additionally, our study is a non-survival model, in contrast to their survival model. The necessity of splenectomy in our porcine hemorrhage model was thoroughly discussed with our Scientific Officers at the DoD Institute of Surgical Research (ISR), and we ultimately agreed to implement pre-injury splenectomy in our studies.

Given these substantial differences in hemorrhage dynamics and experimental endpoints, the applicability of their conclusions on splenectomy to our model is limited. As noted in the referenced study, the necessity of splenectomy is model-dependent and influenced by factors such as hemorrhage rate, blood loss volume, and MAP endpoints. The authors explicitly caution against extrapolating their findings to other models, particularly uncontrolled hemorrhage models like ours, where physiological responses may differ significantly due to the severity and rapidity of blood loss.

While we acknowledge that splenectomy could potentially influence hemostatic and inflammatory responses, determining whether splenectomy is essential in our specific model requires further evidence. Future experiments are needed to assess the impact of splenectomy in uncontrolled hemorrhage models and to ascertain whether it can be omitted without confounding the results. This would help establish whether the spleen's contribution to the physiological response under such conditions justifies the procedure. We have included this information as a limitation in the Discussion section. **(See revised manuscript, Page 17)**

The discussion is as follows: "The necessity of performing a splenectomy in swine hemorrhage models remains a topic of debate. Furthermore, the requirement for splenectomy is highly model-dependent, influenced by factors such as the rate and volume of blood loss, the hemorrhage endpoint, and the specific study objectives.⁴⁸ While splenectomy may not be essential in controlled hemorrhage models with moderate blood loss rates and fixed-pressure endpoints, its role in uncontrolled hemorrhage models like ours, characterized by rapid and severe blood loss, is unclear. Further studies are needed to determine whether splenectomy is necessary in such models and to evaluate its impact on hemostasis, inflammation, and overall physiological responses."

10. How did you measure post-treatment blood loss accurately? Often, the exact amount of blood coming from a junctional area of free bleeding, especially when some material was applied, may be challenging. It is likely that some of the blood in the area was coagulated. Or was it not? The level of accuracy is artificially high. For example, the authors report that "the average blood loss, including the blood absorbed by the hemostasis, was only 42.908 mL". How did you measure the blood loss from a free bleeding model with this accuracy? It is inherently difficult to measure the exact bleeding in free bleeding models. Moreover, even if the .908 mL were truthfully accurate, please discuss the clinical relevance of this accuracy.

Response: We appreciate the reviewer's insightful comments regarding the accuracy of blood loss measurements in a free-bleeding model. The following equation was employed for the quantification of post-treatment blood loss. **(See revised manuscript, Page 33 and Eq. 21)**

$$\text{Post Treatment Blood Loss} = (WC_b - WC_e) + (WG_w - WG_d) + (WM_b - WM_d)$$

Where WC_b represents the weight of the canister used to collect blood lost after treatment until death or after 180 min of observation, and WC_e denotes the empty weight of the canister. WG_w is the weight of wet gauze used for post-treatment manual compression for 3 min, and WG_d is the dry weight of the gauze. WM_b

represents the weight of application materials (hemostats) after treatment, and WM_d is the weight of the hemostats before treatment.

We reported the calculated mean values in the Results Section. In the main manuscript, we have revised the reported blood loss value to 42.9 mL for clarity and consistency. **(See revised manuscript, Page 11)** Additionally, we have updated all blood loss data throughout the manuscript. These changes have been applied consistently across the entire manuscript to enhance clarity and address the concern regarding clinical relevance.

The revision is as follows: “Post-treatment blood loss in the NMA group (0.9 ± 0.2 mL/kg) was significantly lower than in the QCG[®] group (31.2 ± 13.6 mL/kg) and the XStat[®] group (17.5 ± 6.3 mL/kg) **(Fig. 6d)**. The average blood loss, including the blood absorbed by the hemostats, was only 42.9 mL in the NMA-treated group, whereas the XStat[®] and QCG[®] groups had approximately 634.9 mL and 1265.2 mL, respectively **(Fig. 6d)**.”

11. “Additionally, all untreated animals expired”. *This is an unnecessary euphemism. Expired means died.*

Response: We have replaced "expired" with "died" to ensure clarity and consistency in reporting the outcomes of untreated animals. **(See revised manuscript, Page 11)**

The revision is as follows: “Additionally, all untreated animals died from exsanguination between 10 and 45 min, losing 1603.2 ± 343.0 mL of blood after 30 s of free bleeding following the injury **(Fig. 6c, d)**.”

12. *Coagulation tests are missing, and data on how standard coagulation was affected by the treatment.*

Response: We appreciate the reviewer’s observation regarding the inclusion of coagulation test data. We have provided a detailed overview of coagulation parameters in swine experiencing non-survivable junctional hemorrhage following the administration of the tested samples. This includes data on how the treatments affected hematologic values, arterial blood gas values, blood pH, body temperature, and standard coagulation metrics, such as prothrombin time, activated partial thromboplastin time, INR, and fibrinogen levels. **(See revised manuscript, Page 13 and Supplementary Tables 20-24)**

The revision is as follows: “Additionally, the NMA group demonstrated the consistent level of hemoglobin, hematocrit, platelets, and lactate throughout the study period **(Supplementary Table 20)**. In contrast, the hemoglobin level in the XStat[®] group decreased to 7.94 ± 1.47 from baseline due to significant blood loss before achieving hemostasis **(Supplementary Table 20)**. No discrepancies were observed in PT, aPTT, INR, and fibrinogen values in the NMA group **(Supplementary Table 21)**. In contrast, the aPTT in the XStat[®] group increased by at least three times, contributing to unstable hemostasis. Arterial blood gas parameters, including HCO_3 , PaO_2 , $PaCO_2$, and $EtCO_2$, remained stable in the NMA-treated group and the surviving subjects in the XStat[®] group **(Supplementary Table 22)**. There was no significance in blood pH and body temperature between the XStat[®] and NMA groups **(Supplementary Tables 23, 24)**.”

Reviewer #5 (Remarks to the Author):

Remarks to the Author: I co-reviewed this manuscript with one of the reviewers who provided the listed reports. This is part of the Nature Communications initiative to facilitate training in peer review and to give appropriate recognition to Early Career Researchers who co-review manuscripts.

Response: We thank Reviewer #5 for his/her involvement in the review process and for co-reviewing this manuscript. We appreciate the effort and time invested in this process.

Response to Referees

We have thoroughly revised the manuscript based on the editorial request. The revised section was highlighted in yellow.

No new comments from reviewers.

1. In Fig1 f, the figure are not in the same scale bar (NA, MA and NMA group).
2. In Fig2 all the measurement need see the model and the real membrane.
3. Fig3. Need put the real membrane shape and change in the Fig,.
4. In Fig4 m, the figure is not in the real shape,
5. Fig5 a. why the is no different in the NA group, MA group and NMA group?
And why the clotting time is different in NA group, MA group and NMA group?
The Fig5 h need to put in the position of Fig5a.
6. in line 217-220, why the author did not chose the same long and size of the Xstat and NMA materials.
7. In line 379-381 , “The aerogel demonstrated 380 excellent mechanical properties, including a high specific elastic modulus, high resilience, rapid shape recovery, and exceptional robustness” , the author need to put the real comparation data of the high specific elastic modulus, high resilience, and rapid shape recovery with other group.
- 8 . In line 411-419, the hemostasis mechanism is complicate, it need blood coagulation factors, not just RBCs and platelets accumulate, pls put more evidence for the coagulation procedure of NMA materials.
9. In line 431-432, the author use the recent report data, the condition is changed, the author need to do the experiment in the same condition, so the data of control, XStat®, and QCG® groups from the recent report need to make sure.